# SNARE disassembly requires Sec18/NSF side loading

Yousuf A. Khan [1,2,3,4 ✉], K. Ian White [1,2,3,4,5], Richard A. Pfuetzner[1,2,3,4,5], Bharti Singal [6], Luis Esquivies[1,2,3,4,5], Garvey Mckenzie [7], Fang Liu [7], Katherine DeLong [1], Ucheor B. Choi[1,2,3,4,5], Elizabeth Montabana[6], Theresa Mclaughlin [7], William T. Wickner[8] & Axel T. Brunger [1,2,3,4,5 ✉]

SNARE (soluble *N*-ethylmaleimide-sensitive factor (NSF) attachment protein receptor) proteins drive membrane fusion at different cell compartments as their core domains zipper into a parallel four-helix bundle. After fusion, these bundles are disassembled by the AAA+ (ATPase associated with diverse cellular activities) protein Sec18/NSF and its adaptor Sec17/α-SNAP to make them available for subsequent rounds of membrane fusion. SNARE domains are often flanked by C-terminal transmembrane or N-terminal domains. Previous structures of the NSF–α-SNAP–SNARE complex revealed binding to the D1 ATPase pore, posing a topological constraint as SNARE transmembrane domains would prevent complete substrate threading as suggested for other AAA+ systems. Using mass spectrometry in yeast cells, we show N-terminal SNARE domain interactions with Sec18, exacerbating this topological issue. We present cryo-electron microscopy (cryo-EM) structures of a yeast SNARE complex, Sec18 and Sec17 in a nonhydrolyzing condition, which show SNARE Sso1 threaded through the D1 and D2 ATPase rings of Sec18, with its folded, N-terminal Habc domain interacting with the D2 ring. This domain does not unfold during Sec18/NSF activity. Cryo-EM structures under hydrolyzing conditions revealed substrate-released and substrate-free states of Sec18 with a coordinated opening in the side of the ATPase rings. Thus, Sec18/NSF operates by substrate side loading and unloading topologically constrained SNARE substrates.

Cellular compartmentalization, growth, hormone secretion, transport, neurotransmission and many other pathways depend on precise, rapid and regulated membrane fusion[1,2]. Membrane fusion in eukaryotic cells is mediated by a highly conserved superfamily of SNAREs (soluble *N*-ethylmaleimide-sensitive factor (NSF) attachment protein receptors)[3]. All SNAREs share a characteristic 60–70-aa SNARE domain often flanked by a C-terminal transmembrane domain, membrane anchors and a folded N-terminal variable domain specific to the SNARE's function and intracellular pathway[4]. SNAREs on opposing membranes interact primarily through their SNARE domains to form a parallel *trans*-SNARE complex, juxtaposing two different membranes[5,6]. Membrane fusion commences when these SNARE domains zipper together

in a directed fashion[7,8]. After fusion, the SNAREs form a highly stable parallel helical bundle, the so-called *cis*-SNARE complex[8]. This highly stable four-helix *cis*-SNARE complex is disassembled by Sec18 (NSF in higher eukaryotes)[9–11] to provide the energy for subsequent membrane fusion events[12].

Sec18/NSF is a universally conserved AAA+ (ATPase associated with diverse cellular activities) protein translocase[13–15]. Sec18/NSF consists of an N domain, an active D1 AAA+ domain and a catalytically inactive D2 AAA+ oligomerization domain[16]. It was initially discovered as a critical complementation group required by the yeast secretory pathway and it is now known for its role in recycling SNAREs[17] and SNARE assembly quality control for proper SNARE complex assembly[18–21]. Sec18/NSF,

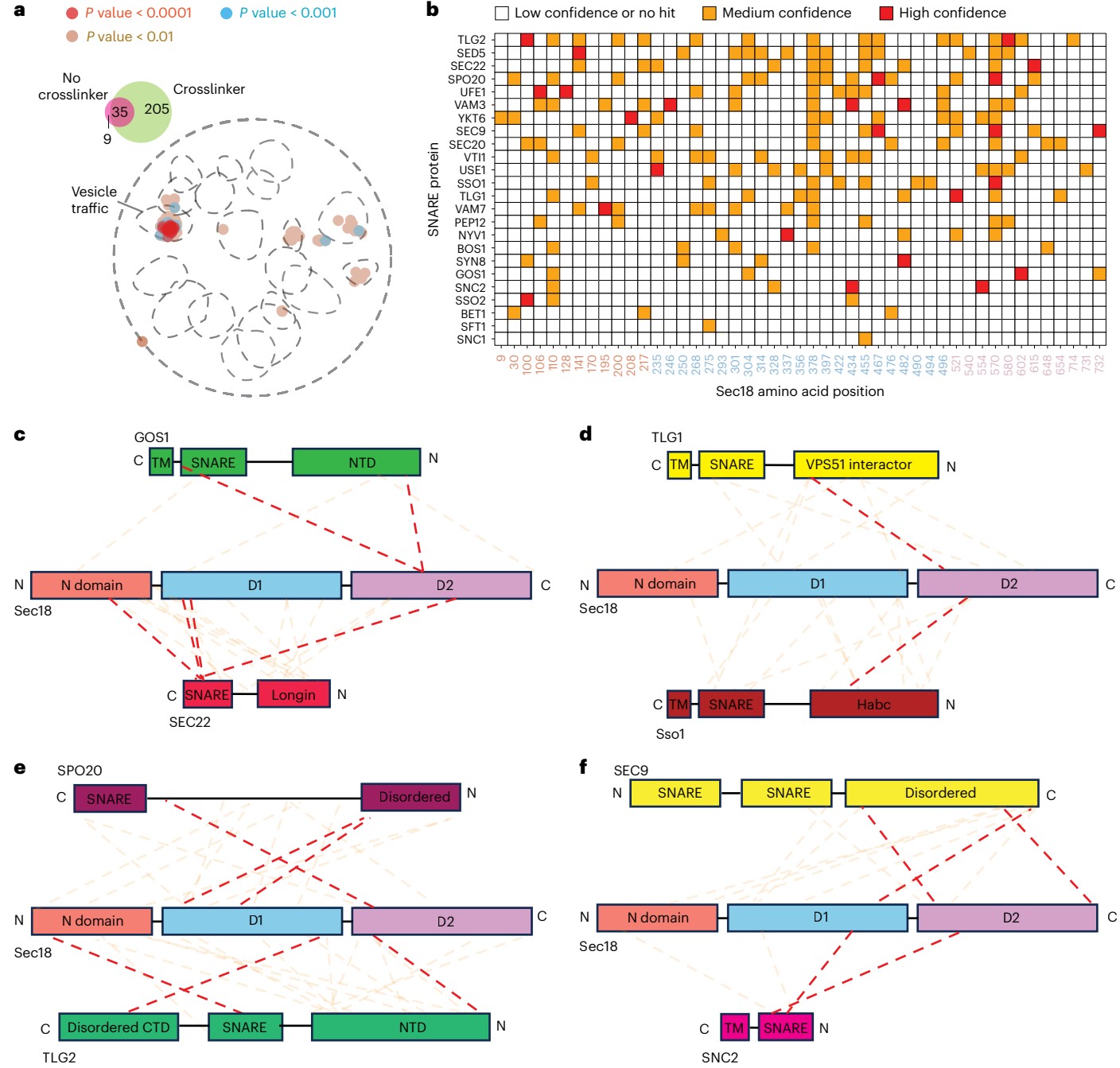

**Fig. 1 | In vivo XL-MS of Sec18. a**, Top left, Venn diagram of proteins found in the condition without a crosslinking agent (red) added and with crosslinking agent added (green). Bottom right, SAFE analysis[32] of yeast proteins that specifically crosslink to Sec18. Blobs represent enrichments in the pathway. Blobs of different colors represent enrichments that are specific to different *P*-value thresholds (brown, <0.01; blue, <0.001; red, <0.0001). *P* values were derived from Fisher's exact test with Bonferroni correction for multiple testing. **b**, Heat map of yeast SNARE proteins crosslinking to specific Sec18 positions. The Sec18 primary sequence along the *x* axis is color-coded by domain and yeast SNARE proteins with at least one medium-confidence crosslink are shown along the *y* axis. Red squares represent positions where one or more manually verified, high-confidence crosslinks were found (Methods). Orange squares represent where one or more manually verified medium-confidence crosslinks were found. White squares represent positions where manually verified crosslinks were not found or were of low confidence. **c–f**, Crosslinking schematic of Sec18 and SNAREs that contain at least one high-confidence crosslink to the Sec18 D2 domain. Red dashed lines represent high-confidence crosslinks and orange dashed lines represent medium-confidence crosslinks.

with the adaptor protein Sec17 (α-SNAP in higher eukaryotes), disassembles *cis*-SNARE complexes. The disassembly requires multiple ATP hydrolysis events in the presence of Mg$^{2+}$; as few as six ATP hydrolysis events are sufficient[22,23].

Cryo-electron microscopy (cryo-EM) structures of the mammalian 20S complex (NSF, α-SNAP and neuronal SNARE ternary complex) in a nonhydrolyzing condition (that is, in the absence of divalent cations) revealed a supramolecular architecture in which the NSF N domains bind α-SNAP molecules, three or four of which in turn bind the four-helix SNARE complex consisting of the SNARE domains of syntaxin 1A, SNAP25 and synaptobrevin[10,24]. The N-terminal residues of the SNARE domain of SNAP25 were bound to the D1 ring pore without apparent ATP hydrolysis, where it interacts with several conserved tyrosine amino acids in a spiral staircase-like pattern[10,13]. In these EM

maps, no ordered density was observed in the D2 ring pore. The D2 ring is catalytically inactive and primarily responsible for NSF oligomerization rather than substrate engagement[25]. The interaction between the SNARE substrate and the D1 pore is like that observed for other AAA+ translocases and suggests a conserved mechanism for substrate threading through the D1 pore. However, the membrane anchors and domains of the SNAREs would seemingly prevent complete threading of the type suggested for other AAA+ systems[26–28], posing a topological challenge.

Furthermore, SNAREs often contain globular N-terminal domains of variable length and structure; for example, the N-terminal domain of syntaxin consists of a three-helix bundle (Habc domain) involved in regulating its function[29] and the N-terminal domains of Use1 and Sec20 in part form a stable 255-kDa tethering complex[30]. Complete threading would imply that such N-terminal domains are somehow unfolded. These topological constraints surrounding SNARE loading, processing and release through Sec18/NSF are further compounded by the observation that Sec18/NSF disassembles all SNAREs in all cellular contexts[3,4,31], which all contain a variety of different N-terminal and C-terminal domains and membrane arrangements and linkages.

## In vivo crosslinking mass spectrometry (XL-MS) with Sec18

These questions led us to investigate the space of Sec18/NSF–SNARE interactions through in vivo protein XL-MS in yeast. Because of its power as a model system and relatively simple SNARE proteome, *Saccharomyces cerevisiae* is an excellent model for investigating Sec18 interactions with different substrates in live cells. As such, we developed a protocol for in vivo XL-MS in yeast to identify binding partners of Sec18 (Extended Data Fig. 1a and Methods).

In total, 35 identified proteins were shared between disuccinimidyl glutarate (DSG) crosslinker-treated and untreated conditions (that is, they were nonspecific proteins) (Fig. 1a). The nine proteins observed only in the untreated condition had few unique spectra mapping to their identified proteins, suggesting that they were not detected in the treated condition because of their low abundance and difficulty of consistent detection. The 205 proteins unique to the 5 mM DSG-treated condition were processed using spatial analysis of functional enrichment (SAFE) to visualize the various cellular processes to which the identified proteins contribute[32]. At increasing levels of significance thresholds ($P = 0.01$, $0.001$ and $0.0001$), the identified proteins only enriched the vesicle-trafficking processes. Gene ontology analysis also revealed lesser-known processes, such as vacuolar acidification and ergosterol biosynthesis (Supplementary Fig. 1), consistent with Sec18's functions in these contexts[33–35]. The enrichment of these processes validated our crosslinking protocol for specifically targeting Sec18 and its binding partners inside the cell.

Within our enriched protein dataset, we found many yeast SNAREs involved in different pathways and compartments of the cell. We then mapped crosslinked residues of these SNAREs to those of Sec18 (Fig. 1b). Using a series of empirical constraints (Methods), we classified a crosslink as high confidence if it met all constraints, medium confidence if it met some but not all and low confidence if it did not. Because crosslinking was performed in live cells, we expect these crosslinks to represent interactions between SNAREs and Sec18 during substrate loading, disassembly and substrate release.

Previous cryo-EM structures of the complex of NSF, α-SNAP and neuronal SNAREs in a nonhydrolyzing condition revealed that the four-helix SNARE bundle interacts with between two and four α-SNAP molecules, which in turn are bound by the N domains of NSF; the N-terminal end of one of the SNAREs is bound to the pore of the D1 ring[10,24]. Consistent with these structures, we found high-confidence crosslinks between SNARE domains and the N domain or D1 domain (examples in Fig. 1c,e,f). Additionally, we observed several high-confidence crosslinks of yeast SNARE proteins to the D2 domain, an unexpected result given an absence of

SNARE density in the D2 ring in these previous cryo-EM structures and that the D2 domain is catalytically inactive[16] with no previously reported role in SNARE recycling. Considering this unexpected result, we thus focused on SNAREs with at least one high-confidence crosslink to the D2 domain of Sec18 (Fig. 1c–f).

Specifically, seven of ten high-confidence D2 crosslinks connect to regions N-terminal to a SNARE domain (Fig. 1c–f). For example, for Sso1, the high-confidence crosslink to D2 involves its Habc domain (Fig. 1d and Extended Data Fig. 1b). The Habc domain is a stable three-helix bundle[29]; thus, assuming complete threading through the D1 and D2 pores, the Habc domain (~30 Å in diameter) would have to be unfolded transiently. The remaining three high-confidence D2 crosslinks connect to C-terminal regions within SNARE domains. At first glance, this suggests complete substrate threading through both the D1 and D2 rings, akin to other AAA+ protein translocases[36]. However, considering the topology imposed by membrane domains, anchors and folded N-terminal domains, how is a SNARE substrate loaded and released? To answer these vexing topological questions, we next determined cryo-EM structures of this orthologous yeast complex together with Sec18 and Sec17.

## The Sec18–Sec17–Sso1–Snc1–Sec9 (y20S) complex

To corroborate the surprising liquid chromatography (LC)–MS/MS results, we determined cryo-EM structures of a yeast SNARE complex together with Sec18 and Sec17. We chose the Sso1–Sec9–Snc1/Snc2 complex (referred to as the ySNARE complex) because each component of this complex crosslinked with high confidence to the D2 domain of Sec18 (Snc1 is highly homologous to Snc2). We prepared the y20S complex in a nonhydrolyzing condition (Extended Data Fig. 2 and Methods) and, after single-particle cryo-EM data collection and processing, we obtained 381,591 high-quality particles that yielded eight three-dimensional (3D) classes into which models were built (Fig. 2a–e, Supplementary Fig. 2a,b and Extended Data Table 1).

In the structures of all eight classes, Sso1 is threaded through the D1 pore of Sec18 (Fig. 2a). A characteristic phenylalanine side-chain density in Sso1 is present at the same position relative to Sec18 in all classes, allowing for reliable indexing of Sso1. The nucleotide states and the arrangements of D1 around the substrate were also the same among all eight classes, with the ADP-engaged E protomer forming the top of a spiral staircase of tyrosine residues. Protomers D, B and C were all ATP bound and formed the middle of the staircase. The bottom protomer, A, formed the base of the staircase. Protomer F, with no apparent density for nucleotide in D1, was not associated with Sso1 (Fig. 2b and Extended Data Fig. 3a). This arrangement of the protomers is driven, in part, by the angle between the more mobile D1 domain and rigid D2 domain. From protomers E to A, the angle between the D1 and D2 decreases monotonically (Fig. 2c). The protomer at the top of the staircase maintained the largest angle between D1 and D2, while the one at the bottom had the smallest. The eight classes from the nonhydrolyzing y20S dataset also showed differences in the ySNARE–Sec17–Sec18 N domain subcomplex arrangement ('spire') and Sec17 stoichiometry above the D1 ring of the y20S complex before disassembly (Figs. 2d,e and 3a,b and Supplementary Discussion).

The high-confidence crosslinks for Sec18 intraprotomer and interprotomer interactions are consistent with the cryo-EM structure, specifically with the arrangements of the N domain and D2 domain (Fig. 3c). The high-confidence crosslink between the D1 domain of Sec18 and Snc2 is also consistent with our cryo-EM structures; the D1 domain is proximal to the SNARE domain of Snc2 in the initial loading state of the 20S complex where Sso1 is loaded into the D1 pore. The other high-confidence crosslink occurs between Snc2 and a D2 domain residue at the interface between the D1 and D2 protomers. However, there is no density for Snc2 in our cryo-EM structures in the D1 pore, suggesting that these crosslinks may represent a transient

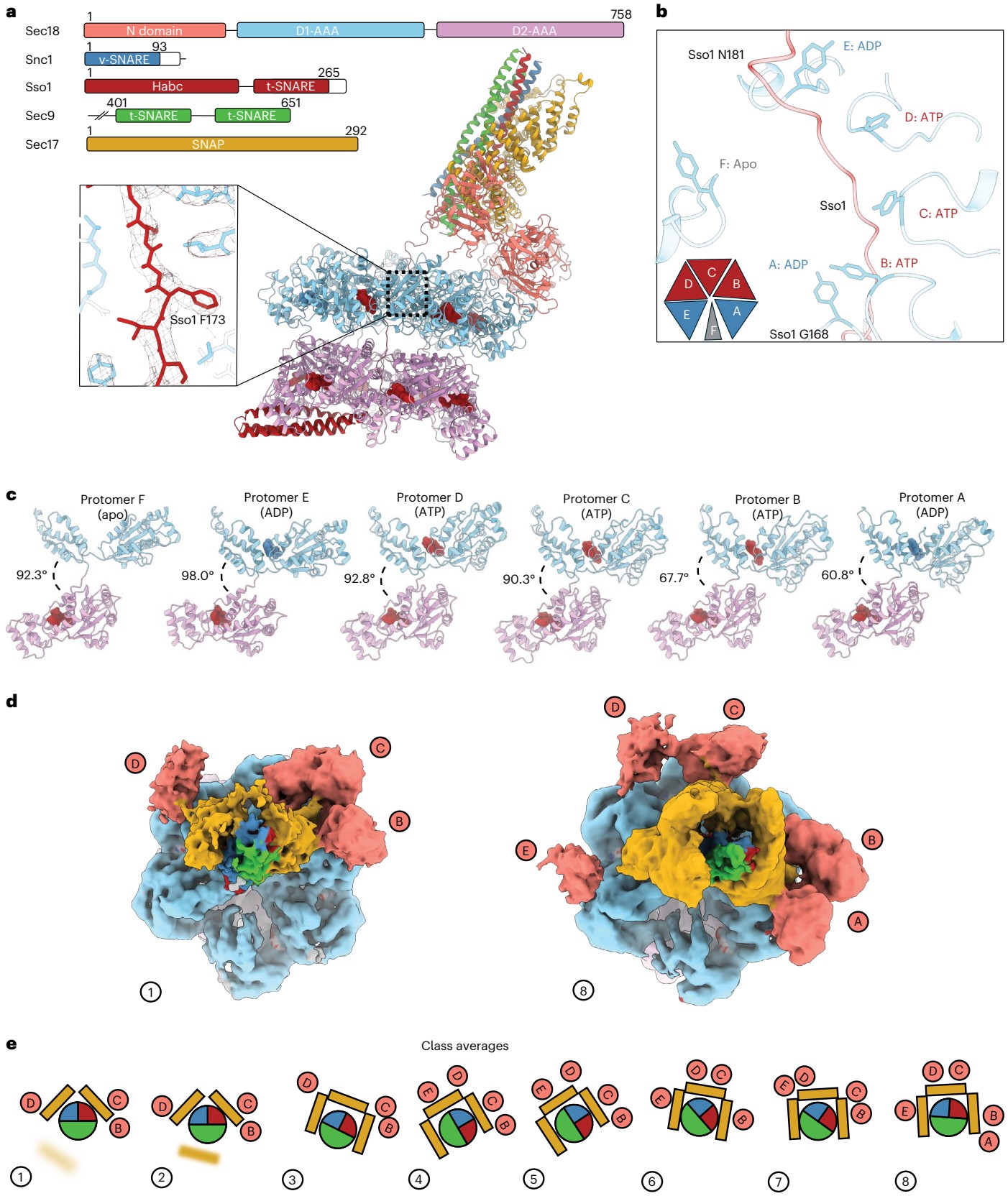

**Fig. 2 | The y20S supramolecular complex. a**, Top left, domain diagrams of the components used to prepare the y20S complex. This coloring is maintained throughout. Bottom right, the structure of class 1 of y20S, with ATP and ADP nucleotides (red and blue, respectively). Bottom left, cryo-EM density for class 1 and the Sso1 atomic model around residue F173 (inset). **b**, Sso1 substrate is engaged by Y315 in all D1 protomers with bound nucleotide. **c**, Conformations of the protomers of Sec18 when engaged to substrate. **d**, Top-down views of class 1 and class 8 cryo-EM maps. **e**, Cartoon top-down views of eight y20S classes. These classes differ primarily by spire configuration (that is, the pattern of N domain and α-SNAP engagement).

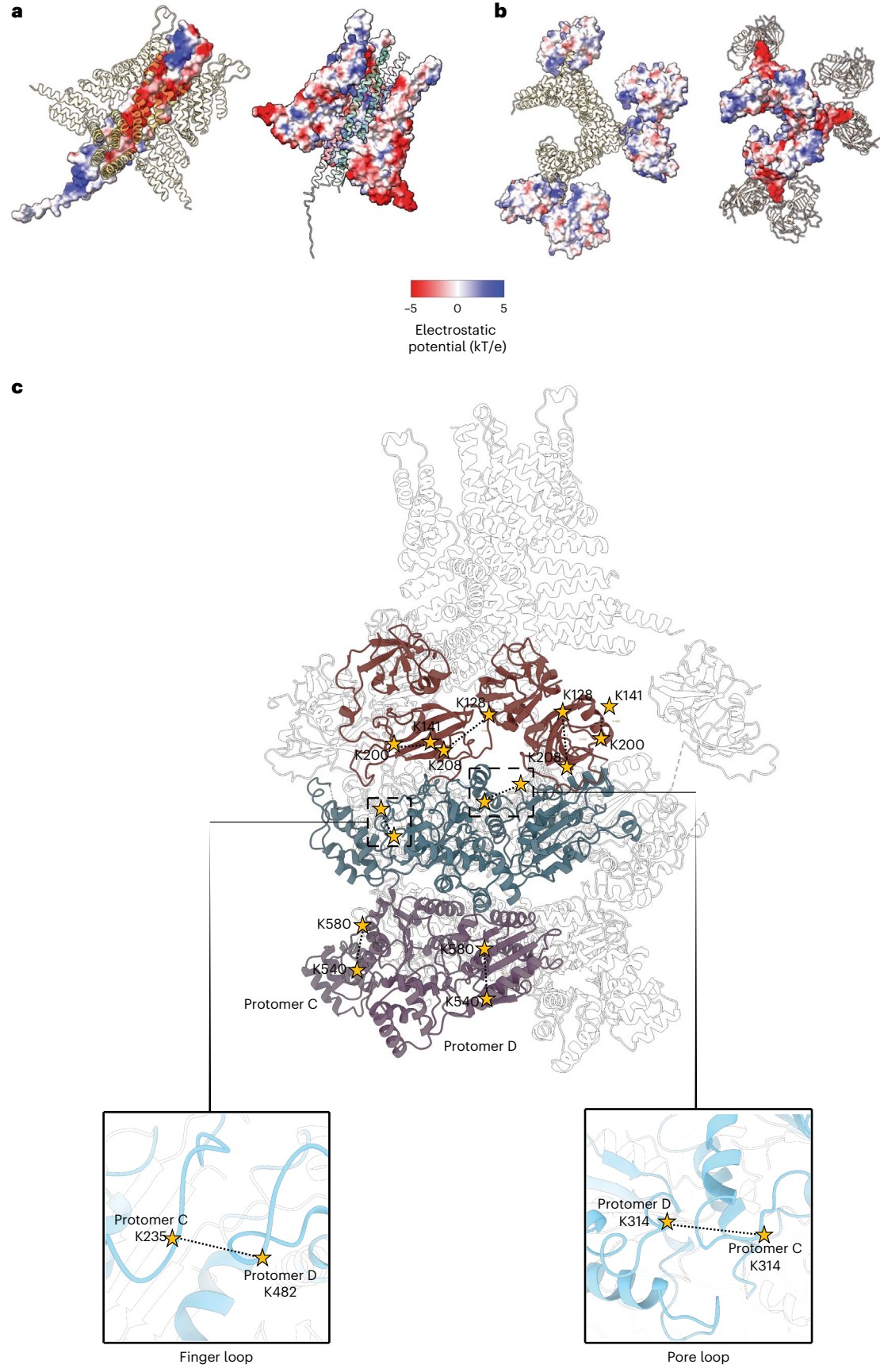

**Fig. 3 | Intraprotomer and interprotomer crosslinks mapped on the y20S complex structure. a**, Electrostatic surface potentials of the interaction between SNAREs and Sec17. Negatively charged regions are colored red and positive ones are colored blue. **b**, Electrostatic surface potentials of the interaction between Sec17 and the N domains of Sec18. Alternating views of these interactions are shown to demonstrate the complementarity of the electrostatic interactions. **c**, For clarity, only two Sec18 protomers of the y20S complex are shown. The black dashed lines represent crosslinks designated as high quality and superimposed over the diagram. The two insets show close-up views of two interprotomer crosslinks.

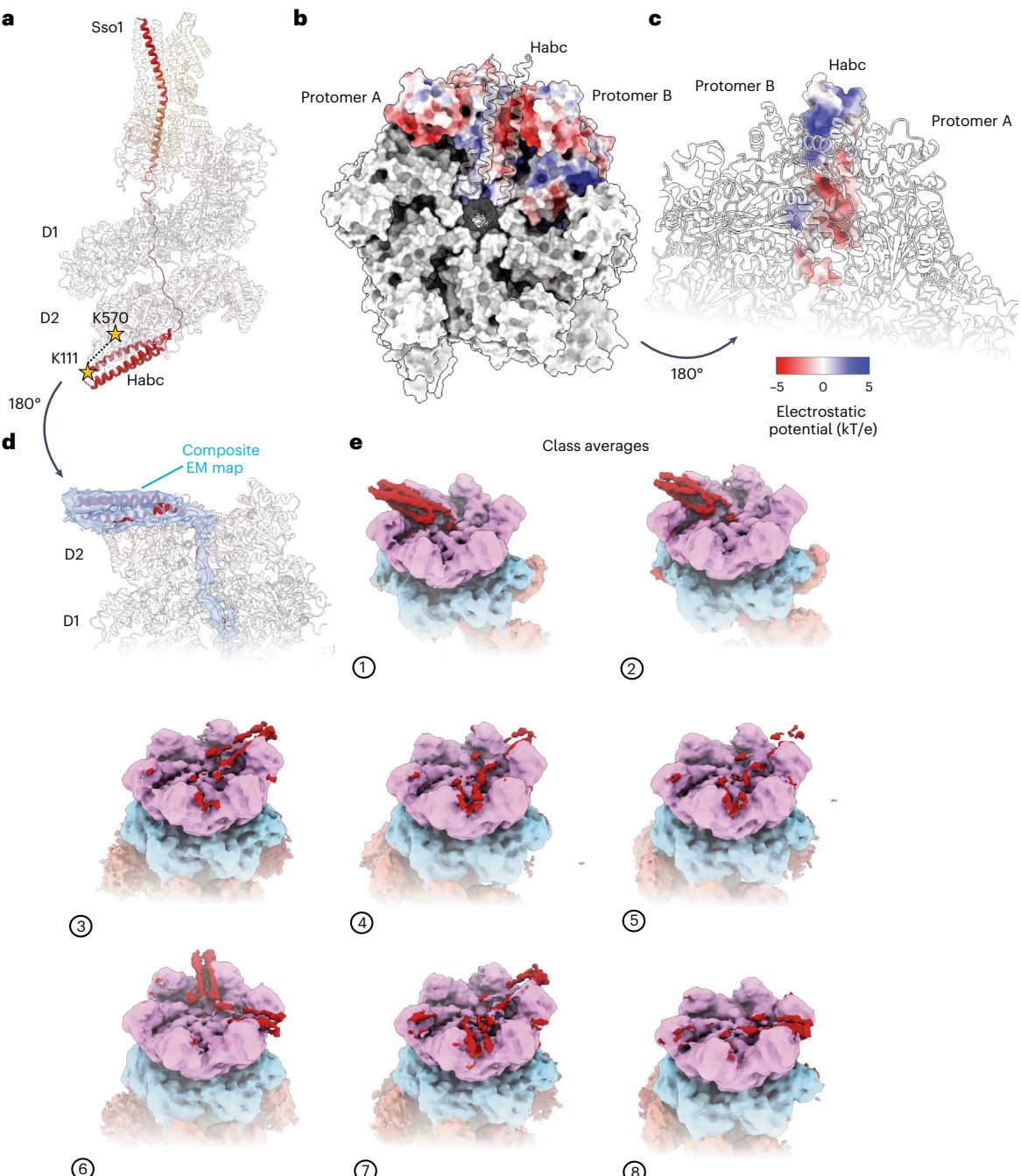

**Fig. 4 | Sso1 is threaded through the D1 and D2 domains. a**, y20S class 1 atomic model emphasizing the position of Sso1. Stars represent the crosslink identified by XL-MS (Fig. 1b). **b**, Bottom view of y20S class 1 D1 ring; the Habc domain is packed against the D2 ring surface, between protrusions corresponding to the C-terminal helical regions of two D2 small subdomains. All surfaces are colored gray, except for protomers A and B, which are colored by their electrostatic potential **c**, Rotated view of y20S class 1 relative to **b** showing the Habc tucked between protomers A and B, with the Habc domain colored by its electrostatic potential **d**, Composite cryo-EM map (blue, RELION unsharpened map: Habc domain and D1 density; cryoSPARC 3DFlex map: D2 density) of y20S class 1. **e**, Cryo-EM maps of all eight y20S classes reveal the Habc domain associated with the Sec18 D2 ring in all classes (red). In six of eight classes, the Habc density is radially averaged into multiple discrete positions around the D2 ring, leading to more diffuse density.

state. Moreover, interprotomer crosslinks were found between pore loop regions and between the interprotomer loop between residues 476 and 490 and N domain linker regions of D1. These high-confidence crosslinks were within the theoretical maximum crosslinking distance ($N_\zeta$–$N_\zeta \approx 40$ Å)[37].

Remarkably, our cryo-EM structures revealed that Sso1 is loaded into both the D1 and the D2 pores because the Habc domain of Sso1 interacts with the outside of the D2 ring, with class 1 having the most well-defined, discrete density (Fig. 4a–e and Extended Data Fig. 3b). In class 1, the density for Sso1 within Sec18 is continuous from the D1 pore entrance to the D2 pore exit, where the Habc domain begins (Fig. 4a,d). The three-helix bundle of the Habc domain is then positioned outside the D2 pore, where it packs between two small D2 subdomains (protomers A and B) in an interaction driven by charge complementarity. Indeed, the electrostatic potential surface of the Habc domain forms a roughly opposite that of the bottom of the D2 surface against

which it packs (Fig. 4b,c). Given that this charge distribution is replicated between each pair of D2 subdomains around the D2 surface, it is unsurprising that the Habc domain adopts multiple corresponding rotational states over the eight classes, presumably because of the flexible linker that follows it (Fig. 4e). This structural observation is consistent with XL-MS data showing a crosslink between the Habc and the D2 domain of Sec18 (Fig. 4a and Extended Data Fig. 1b). Although the putative distance between the crosslinked residues (K570 in Sec18 and K111 of Sso1) observed in the model for class 1 (42 Å) is at the limit for a crosslinking distance, the conformational flexibility of the Habc domain to rotate and exist in multiple conformations about the D2 ring (Fig. 4e) likely produces conformations well within the crosslinking distance threshold.

## Sec18 does not unfold the Sso1/syntaxin Habc domain

Given that this complex was assembled in nonhydrolyzing conditions, we next asked how Sso1 is loaded into Sec18 given the diameter of the Habc domain of Sso1 is ~30 Å, whereas the D1 pore of Sec18 has a diameter of ~11 Å. In other AAA+ translocases such as ClpX, ATP hydrolysis drives complete substrate threading through the AAA+ pore[26]. However, in the case of Sec18 in a nonhydrolyzing condition, such complete threading would require the Habc domain to unfold without any energetic input from Sec18.

We used two orthogonal approaches to test whether the Sso1 Habc domain unfolds during SNARE disassembly. In our first approach, we used a single-molecule fluorescence resonance energy transfer (smFRET) system developed previously for NSF, α-SNAP and neuronal SNAREs[11] in which individual SNARE domains were synthetically linked and stochastically labeled with FRET pairs. This system allows one to observe multiple rounds of SNARE disassembly as the disassembled SNARE domains readily reassemble into a *cis*-SNARE complex because of the covalent linkages. We chose this system because it is well established. Moreover, it is relevant for the Sec18 system studied here because the primary sequence is conserved in the core regions of Sec18/NSF responsible for ATP hydrolysis and substrate processing (Extended Data Fig. 4a). In addition, the D1 and D2 rings are structurally conserved when comparing the D1 and D2 rings of Sec18/NSF with engaged substrates, with an average Cα root-mean-square deviation (r.m.s.d.) of 1.64 Å (Extended Data Fig. 4b,c). The only protomer that exhibits much difference is the F protomer, likely because of its mobility and not because of inherent differences between the structures. Furthermore, we tested the ability of Sec18 and Sec17 to disassemble fluorophore-labeled neuronal SNARE and, vice versa, NSF and α-SNAP to disassemble fluorophore-labeled exocytic ySNARE complex (Extended Data Fig. 5a–e and Supplementary Discussion). We found that both could process the species-ortholog SNARE complex, albeit at different rates, further corroborating the interchangeability of NSF and Sec18.

We, thus, used the smFRET assay to study the effect of NSF on the conformation of the three-helix bundle Habc domain of syntaxin during disassembly. We either stochastically labeled the SNARE domains or stochastically labeled the Habc domain at two distinct residue positions to monitor either SNARE disassembly or the folded state of the Habc domain (Fig. 5a). As in our previous work[11], we observed repeated rounds of disassembly and reassembly for the SNARE domains as indicated by the changes in single FRET intensity over tens of seconds (Fig. 5b,c). In stark contrast, we did not observe a change in single-molecule FRET intensity for the labeled Habc domain in nondisassembly and disassembly conditions, producing a FRET intensity distribution consisting only of a high-FRET state (the small peak at 0 is because of traces where acceptor photobleaching occurred because the particular traces never transitioned back to high FRET and no change in the peak intensity between nondisassembly and disassembly conditions was observed) (Fig. 5d–f). This result suggests that the Habc domain does not unfold during NSF activity.

To corroborate this finding with an orthogonal approach for the yeast system, we double-crosslinked the Habc domain of Sso1 by incorporating an unnatural amino acid, 4-azido-L-phenylalanine[38] (Extended Data Fig. 5f,g). This double crosslink was introduced to prevent the Habc domain from unfolding during any Sec18-driven threading. After crosslinking, confirmed by MS in which we did not detect any uncrosslinked sample (Extended Data Fig. 5g), we performed a disassembly assay on fluorescently labeled uncrosslinked and crosslinked complexes and monitored progress by native gel electrophoresis (Fig. 5g). The top band, representing the fully assembled exocytic SNARE complex, was present in both conditions but disappeared when adding Mg²⁺ to initiate hydrolysis and the disassembly reaction. There was no qualitative difference in the disassembly between uncrosslinked and crosslinked conditions after normalizing for labeling efficiency (Fig. 5h). Thus, the folded state of the Habc domain does not change during yeast SNARE disassembly and crosslinking the Habc domain does not affect the disassembly kinetics. These results for both yeast and neuronal systems further argue against a threading model of Sso1 engagement because the Habc domain would be unable to pass through the D1 and D2 domains.

## Substrate-released Sec18 reveals coordinated ring opening

Together, these observations beg the question of how Sso1 enters the Sec18 ATPase rings. To address this topological challenge, we next determined structures of the y20S complex after initiating Sec18 hydrolysis by adding Mg²⁺ (referred to as the 'hydrolyzing condition'). Informed by a fluorescent protein disassembly assay (Extended Data Fig. 5h), we initiated the disassembly reaction and waited 7 s before sample vitrification.

In the resulting cryo-EM dataset, we found a small number of particles (6,356 or 2.26% of final particles, class 1) that consisted of

**Fig. 5 | Sec18/NSF does not unfold the Habc domain. a**, Diagram of the smFRET disassembly and reassembly assay with NSF/α-SNAP and linked neuronal SNAREs. Either the linked SNARE complex is stochastically labeled (residue number 249 in syntaxin and residue number 82 in synaptobrevin) or the syntaxin Habc domain is stochastically labeled (residue numbers 35 and 105 in the Habc domain). **b**, Representative time trace of labeled linked SNARE complex in the nonhydrolyzing condition. Red represents the acceptor and green represents the donor dye fluorescence intensity time traces. The black line represents fitting by a vbGMM (*n* = 53). Blue represents the FRET efficiency. The right sub panels are the respective probability distributions. **c**, Representative time trace of labeled SNARE complex in disassembly conditions. Red represents the acceptor and green represents the donor dye fluorescence intensity time traces. The black line represents fitting by a vbGMM[51] (*n* = 114). Blue represents the FRET efficiency. The right sub panels are the respective probability distributions. **d**, Representative time trace of labeled Habc domain in the nonhydrolyzing condition. Green represents the donor and red represents the acceptor dye fluorescence intensity time traces. The black line represents fitting by a vbGMM (*n* = 27). Blue represents the FRET efficiency. The right sub panels are the respective probability distributions. **e**, Representative time trace of labeled Habc domain in disassembly conditions. Green represents the donor and red represents the acceptor dye fluorescence intensity time traces. The black line represents fitting a vbGMM (*n* = 22). Blue represents the FRET efficiency. The right sub panels are the respective probability distributions. **f**, Histograms of E-FRET for all four conditions. The blue dotted line represents a Gaussian fit of the data. The *y* axis is the normalized frequency of occurrence of the E-FRET states in the traces. Top left, SNARE E-FRET for the nonhydrolyzing condition; top right, SNARE E-FRET for the hydrolyzing condition; bottom left, Habc E-FRET for the nonhydrolyzing condition; bottom right, Habc E-FRET for the hydrolyzing condition. **g**, Native gel of labeled uncrosslinked or crosslinked yeast SNARE complex labeled with Oregon green maleimide 488. **h**, Quantification of gel densitometry of *n* = 3 independent disassembly experiments. Error bars represent the s.d. of the mean. Densitometries were all normalized to 0-min values.

entire y20S assemblies (Fig. 6a), largely similar to those observed in the nonhydrolyzing condition (Fig. 2). While the resolution of this reconstruction is low (10.88 Å), the D1 ring is flattened relative to reconstructions from the nonhydrolyzing condition (Fig. 6b). Considering that our sample consists of purified y20S complexes before initiating disassembly by adding Mg²⁺, this D1 ring flattening likely occurs because of Mg²⁺ binding or ATP hydrolysis at one or more subunits.

The remaining particles (274,883 or 97.74% of final particles) are substrate free (classes 2–4 in Fig. 6c–e). The D1 ring is also flat in all these substrate-free classes, with all D1 protomers bound to ADP.

This uniform binding to ADP, as opposed to the spiral staircase of apo, ADP-bound and ATP-bound protomers, likely explains the flattened nature of the protomers. In the D2 ring, all protomers are still ATP bound, consistent with their role in oligomerization. Class 4 consists of a configuration where both D1 and D2 rings are heptameric (Fig. 6c) and class 3 is similar to this heptameric class, except that the seventh protomer is only partially occupied (Fig. 6d).

Class 2 has a well-resolved hexameric configuration with a coordinated split in class 2 in the D1 and D2 rings (Fig. 6e). This split spans ~20 Å between protomers and is large enough to accommodate a polypeptide chain entering or leaving the rings. This conformational change

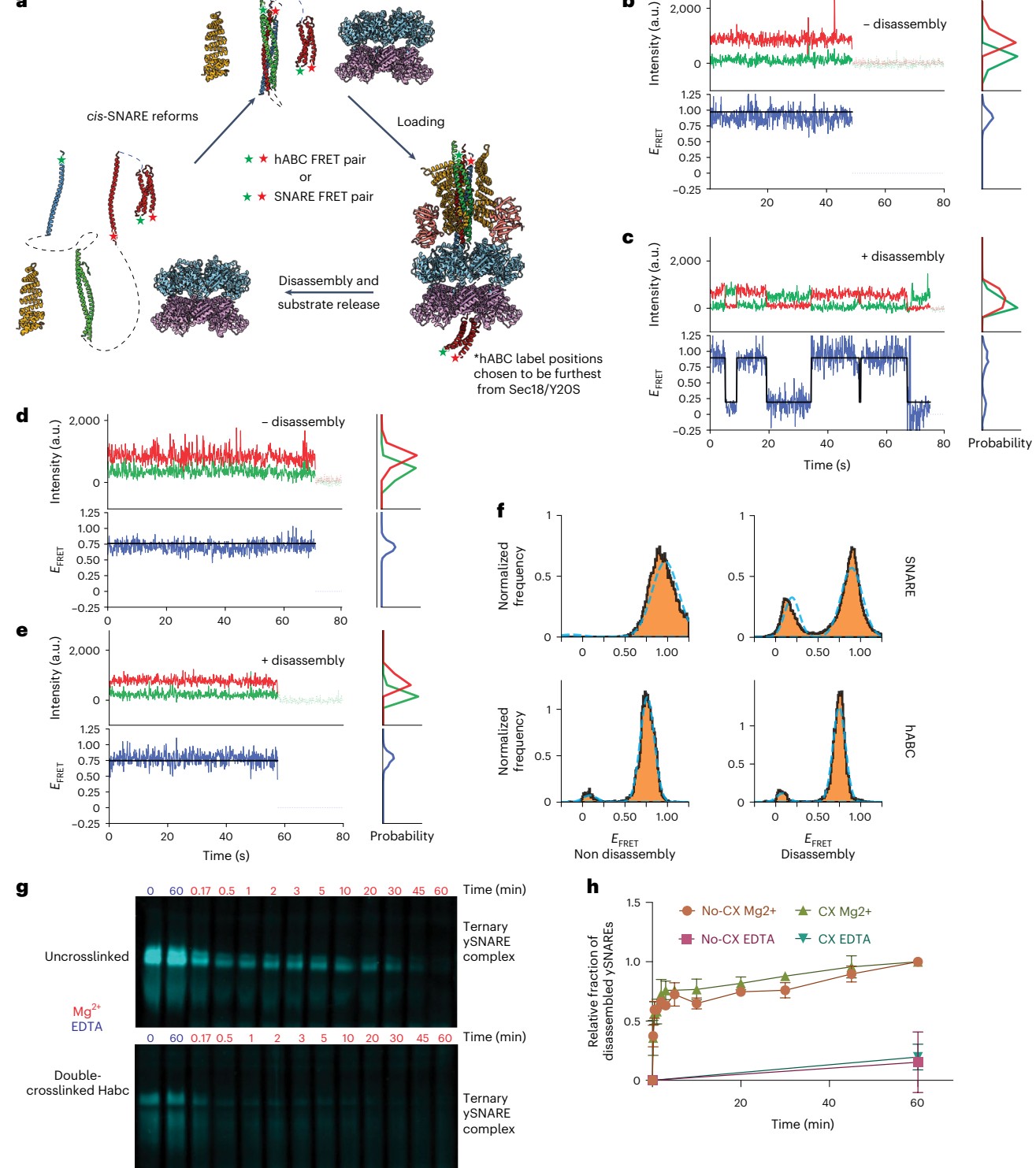

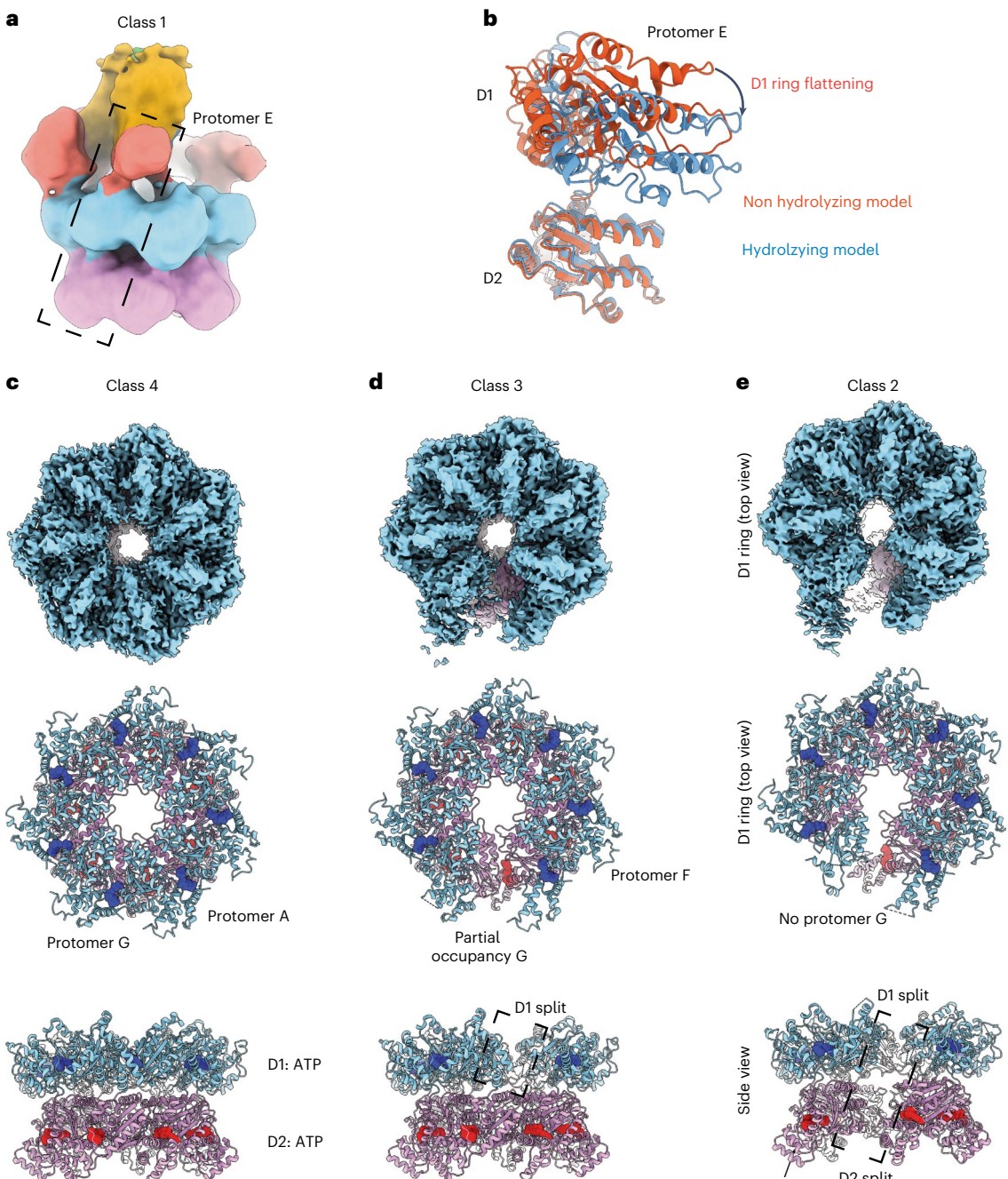

**Fig. 6 | Postdisassembly imaging of y20S reveals substrate-released states.**
**a**, Cryo-EM map of y20S in the hydrolyzing condition, colored by domain.
**b**, The distance between the D1 and D2 domains decreases for some protomers under hydrolyzing conditions, leading to D1 ring flattening. The orange model corresponds to protomer E from the merged D1/D2 focused class in the nonhydrolyzing condition, which was rigid-body fitted into the density of the hydrolyzing condition to generate the blue model. **c**, Cryo-EM map and model of class 4 of Sec18, showing a heptamer. **d**, Cryo-EM map and atomic model of class 3 of Sec18, suggesting a transition state. **e**, Cryo-EM map and atomic model of class 2 of Sec18, showing a split hexamer. In **c**–**e**, the EM map (top) is shown along with atomic models in top (middle) and side (bottom) views.

is likely induced by the nucleotide state and specifically coupled to the presence of ADP throughout the ring following hydrolysis and SNARE substrate processing. The coordinated split in the D1 and D2 domain rings allows the processed SNARE substrate to be released from the side. This side-release mechanism, thus, addresses one part of the topological challenge stated above; threading occurs but the substrate exits from the side, avoiding the membrane domains and linkages in SNARE proteins. We observed similar states, including a coordinate split of the D1 and D2 rings, for the mammalian homolog NSF after processing substrate. This side-release mechanism may also explain the Snc2 crosslink to the residue in the D2 domain that is at the interface between the D1 and D2 protomers (Fig. 1f).

## Substrate-free Sec18 reveals coordinated ring opening

Next, we asked whether this same side opening of Sec18 occurs in the hydrolyzing condition without SNARE substrate. We prepared purified wild-type Sec18 in the presence of ATP and Mg²⁺ in a three-stage procedure that yielded a pure sample used for cryo-EM studies (Extended Data Fig. 6a–c). Two bands were visible in native gels, indicating two

oligomeric states (Extended Data Fig. 6d). Processing and classification led to three classes, two of which appeared to be heptameric and identical (280,935 particles and 73,253 particles, respectively; Fig. 7a) and a third class that contained indeterminate density in the D1 ring (70,513 particles). Density for the N domains was not well resolved in any of the classes, presumably because of the conformational flexibility of the N domains without substrate or adaptor present.

The 3D variability analysis (3DVA)[39] of the third class suggested that it consists of a mixture of heptameric and hexameric Sec18 (Extended Data Fig. 6e), consistent with the two bands seen on a native gel of the sample. The hexameric class reconstructed from 3DVA particle slices from the first mode (Fig. 7b) is like class 2 observed for the y20S complex under hydrolyzing conditions (Fig. 6e), wherein hexameric Sec18 has a large, coordinated split in its D1 and D2 domain rings without substrate. This similarity in Sec18 conformations provides further evidence that the substrate is both loaded and released through the side of the rings, explaining why the Habc domain is not unfolded during Sec18/NSF activity (Fig. 4) and that crosslinking the Habc domain does not substantially affect the disassembly kinetics (Fig. 5h). This solves the second part of the topological question posed above.

The heptameric class (Fig. 7a) is nearly identical to class 4 observed in the y20S complex under hydrolyzing conditions (Fig. 6c). Each protomer in the heptamer is similar to the others, with an average r.m.s.d. < 1 Å between them and the angle between the D1 and D2 rings varying only within a small range of 61.9°–67.1°, in contrast to the hexamer, with a range of 60.8°–92.3° (Fig. 2c and Extended Data Fig. 6f). The D1 pore diameter is 30 Å, larger than that of substrate-bound hexamer, with a diameter of ~11 Å. The substrate-free structures (hexamer and heptamer) are nearly identical (r.m.s.d. = 0.48 Å) to the substrate-released structures.

To test whether these D1 ADP-bound, ring-flattened states of Sec18 observed by single-particle cryo-EM convert to a disassembly-competent state, we tested the same protein preparation used for cryo-EM with a vacuolar–lysosome proteoliposome fusion assay[40,41] (Extended Data Fig. 6h–j and Supplementary Discussion); this assay confirmed that the Sec18 preparation used for cryo-EM was active and processed vacuolar SNAREs. Moreover, NSF adopted this flattened ring state and a heptameric or split hexameric arrangement when exchanged into a buffer containing ATP and $Mg^{2+}$ without substrate (Extended Data Fig. 7 and Supplementary Discussion) and starting from a neuronal 20S complex in a hydrolyzing condition, consistent with our observations that Sec18 and NSF share the same functional mechanism.

## Implications for ATPase activity

The observed structures of Sec18 in substrate-free, substrate-loaded and substrate-released states suggest large conformational change upon ATP hydrolysis, consistent with previous observations[24,42]. In the substrate-free and substrate-released states, examination of the D1 nucleotide-binding pockets reveals critical catalytic residues far from the nucleotide (Fig. 7c). However, in the substrate-engaged state, these conserved residues are tightly bound to nucleotide in a state preceding the binding of divalent cation required to drive hydrolysis. Using the same statistical workflow used to cluster the different y20S classes (Methods), we identified the most significant residues that varied between the no-substrate and substrate-bound states by generating an ensemble of models for each class using ensemble refinement (residues colored red in Fig. 7d and variance contribution per component in Extended Data Fig. 5i). For both the arginine finger loop and the Walker B motif, we observed substantial changes to the side-chain and backbone conformations (Fig. 7e,f), wherein coordinating residues are positioned closer to the nucleotide-binding pocket in the presence of substrate, presumably establishing a hydrolysis-ready state. In the case of the arginine fingers, the side-chain conformational change is likely coupled to nucleotide identity, as hydrolysis disrupts ATP coordination, weakens the interprotomer interface and leads to rigid-body motion

of the D1 large subdomain as the arginine fingers disengage. The case of the shift in the Walker B element, on the other hand, appears to be the flexing of the loop without substantial changes to the sidechains of D349 and E350, induced by the motions of pore loop 2 (GVG motif) that bind to the substrate in the pore.

## Mechanistic details of ring opening

The observed coordinated opening in both D1 and D2 rings in conditions with and without substrate (Figs. 6 and 7) suggests a side-release and side-loading mechanism of substrate. We speculate that the conformations of the split Sec18 observed in these multiple conditions are frequently sampled, enabling a substrate to enter or exit the D1–D2 double ring. To better understand the mechanism driving this opening, we compared the flat D1 closed and open conformations of Sec18 (Fig. 6c,e). The Sec18 protomers are very similar (Extended Data Fig. 8a,b), except for the protomers A and F at the split of the D1 and D2 rings. Rigid-body motions alter the relative positions of these protomers' respective subdomains. We performed a conformational analysis to determine which amino acids contributed most to this conformational change between the open and closed states (Methods); the eight most significant residues contributing to the variance between the two conformations are shown in Extended Data Fig. 8c. These residues are located in flexible regions between the structured subdomains of a protomer (Extended Data Fig. 8d), where changes to the backbone $\phi$ and $\psi$ angles of these residues lead to rigid-body motions of subdomains. For example, four of the eight residues (R409, F410, L418 and N690) are in the linker region between the large and small subdomains of the D1 domain, while G514 is situated between the D1 and D2 domains. These small and large subdomain movements, along with relative motion between the D1 and D2 domains, lead to the split that allows substrate loading or release. Given that this opening is hydrolysis independent and occurs regardless of the presence of substrate, it is likely that Sec18 samples these states upon thermal agitation; subsequently, substrate enters the pore and locks the D1 and D2 rings into an active hexameric form (that is, the 20S/y20S complex), possibly in concert with N-domain engagement.

## Discussion

On the basis of our LC–MS/MS, structural and functional results, we propose a model for SNARE recycling (Fig. 7g), consisting of four phases. Sec18/NSF transiently forms a split-open conformation of both D1 and D2 domain rings in the resting phase with no substrate. Once the *cis*-SNARE complex has been coated with adaptor Sec17/α-SNAP molecules, the Sec18 N domains can bind the subcomplex and position the unstructured linker connecting the N-terminal Sso1 Habc domain to the SNARE domain proximal to the D1–D2 split without unfolding of the Sso1 Habc domain (Fig. 5h), thus promoting the loading of substrate into the catalytic core of the enzyme. More generally, this side-loading mechanism would allow Sec18/NSF to process complexes incorporating all SNAREs in the cell regardless of the presence of folded N-terminal domains, provided that, in the latter case, a long enough linker connects them to the following SNARE domain. Upon forming the (y)20S complex, ATP-driven hydrolysis likely fuels the disassembly of the ySNARE complex by some degree of threading, accompanied by D1 ring flattening. In the case of Sso1, because full threading cannot occur because of the transmembrane domain, partial melting of the N-terminal end of the SNARE domain could be sufficient to destabilize the SNARE complex. Finally, the pore-bound SNARE is released through the side split of both the D1 and D2 rings, regardless of membrane anchors or transmembrane domains in SNARE proteins. We observed a similar coordinated opening of the D1 and D2 rings for the mammalian homolog NSF after processing substrate[43]. The syntaxin Habc domain also does not unfold during NSF-mediated disassembly of the ternary SNARE complex (Fig. 5a–g). The side-loading and side-release mechanism explains how Sec18/NSF can disassemble SNAREs with constrained topologies.

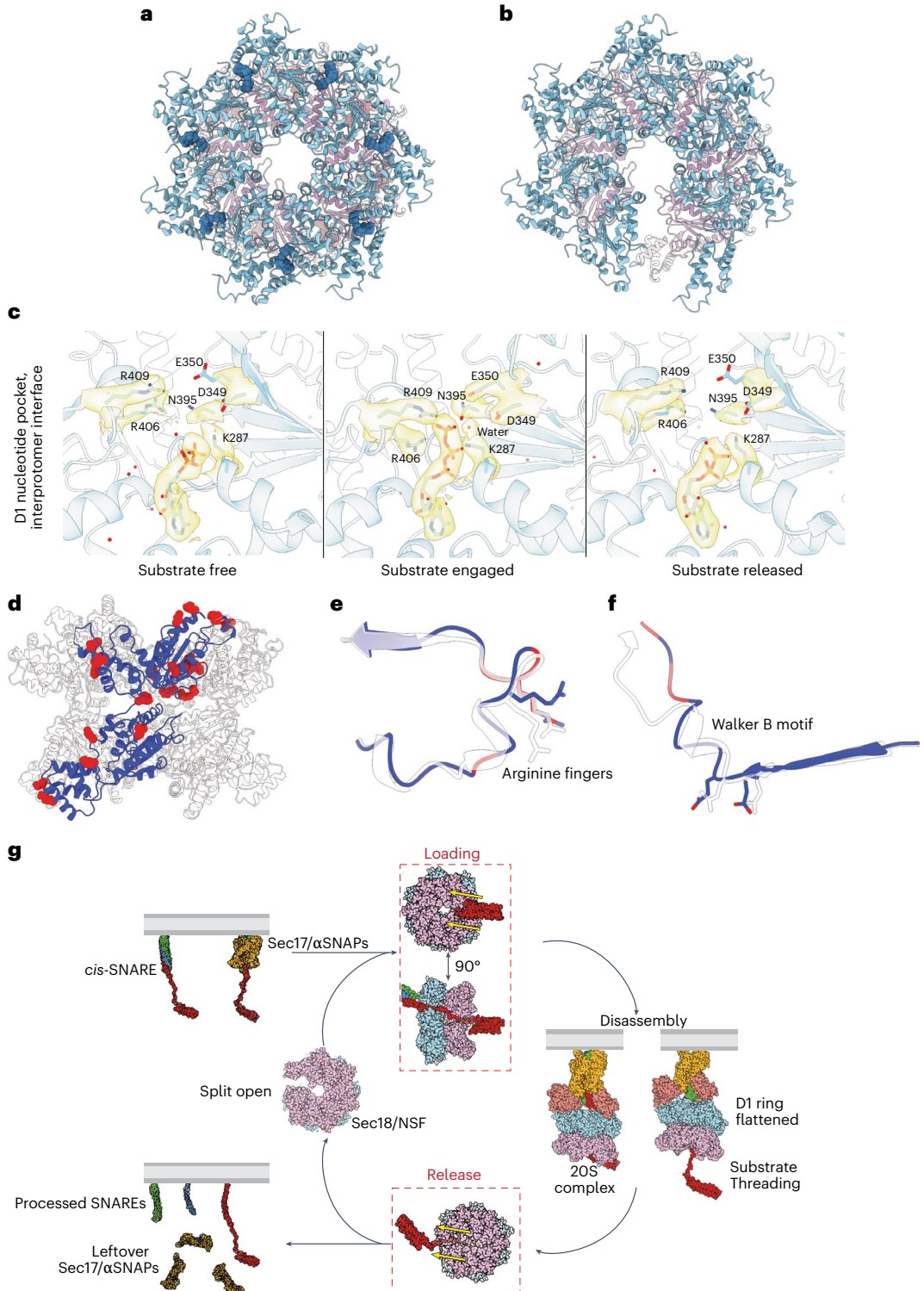

**Fig. 7 | Structures of substrate-free Sec18 in the hydrolyzing condition.**
**a**, Atomic model of the substrate-free heptameric state of Sec18 (class 1). **b**, Atomic model of the split substrate-free hexameric state of Sec18 (derived from 3DVA of class 3). **c**, The D1 nucleotide-binding pocket is remodeled as a function of SNARE substrate binding and nucleotide state. The arginine fingers R406 and R409 are contributed by the neighboring protomer. The substrate-free and substrate-released states are nearly identical in conformation. Water molecules are shown as red spheres. **d**, Atomic model of Sec18 with protomer A colored blue if no significant residue from conformational analysis and colored red otherwise (*P* < 0.05). **e**, Atomic models show the difference of the arginine finger loop in the no-substrate (transparent) and substrate conditions. **f**, Atomic models of the loop proximal to the Walker B motif in the no-substrate (transparent) and

substrate conditions. In **d**–**f**, residues are colored red if they are in the top 5% of residues that significantly vary between conditions. **g**, We propose a general model of SNARE recycling, regardless of the cellular context. First, a *cis*-SNARE complex is coated with at least one Sec17/α-SNAP adaptor molecule, allowing Sec18/NSF to recognize it. The SNARE substrate is then loaded into a split hexamer with a coordinated opening in both D1 and D2 rings through the side, bypassing whatever N-terminal domains may be present. The substrate is then threaded coaxially through the Sec18/NSF pore. Upon completion of processing, the SNARE substrate is released through the side, bypassing the topological constraints of the membrane. Finally, Sec18 returns to its 'resting' state until more substrate is encountered. Yellow arrows indicate the direction of loading and release.

As observed in our LC–XL-MS/MS experiments (Fig. 1), the Sec18 D2 ring can interact not only with the Habc domain of Sso1 but also with other N-terminal accessory domains (such as GOS1, TLG1 and TLG2). In turn, N-terminal domains of SNAREs often interact with other factors. For example, the trimeric Dsl1 complex binds to the SNAREs Use1, Ufe1 and Sec20 through interactions with N-terminal SNARE domains. This 255-kDa complex tethers endoplasmic reticulum membranes together before forming the *trans*-SNARE complex and likely remains complexed before, during and after fusion[30]. Another example is HOPS, a hexameric complex that tethers lysosomal and vacuolar membranes and orchestrates their fusion[44]. HOPS has two Rab-binding subunits for tethering and a SNARE-binding subunit that catalyzes initial SNARE complex assembly[45]. Sec18/NSF and Sec17/α-SNAP cooperate with HOPS to ensure rapid and efficient SNARE-mediated fusion[46,47]. For the case of Dsl1, the N-terminal domains of SNAREs remain in a supramolecular arrangement that must be bypassed. For vacuolar–lysosomal fusion, SNAREs must be preprocessed and handed off to the HOPS complex. These complexes impose topological constraints on both ends of the SNAREs that can be resolved by side loading and side release.

We observed heptameric states of Sec18 (Fig. 6c) and NSF (Extended Data Fig. 7a), split-open hexameric states of Sec18 (Fig. 6e) and NSF and a transition between split hexamer and heptamer of Sec18 (Fig. 6d). Hydrolysis likely accelerates the transition from hexameric to heptameric state through the split-open state for two related reasons. First, while ring splitting itself is hydrolysis independent, hydrolysis is associated with large-scale conformational change in the ATPase rings and likely increases the frequency of sampling the open state. Second, we did not observe the heptamer in a substrate-engaged state. Moreover, it seems unlikely that a heptamer could accommodate substrate given the changes heptamerization induces in the N and D1 layers of the oligomer (an increased diameter would interfere with SNAP recognition of substrate and substrate engagement in the pore). As such, it seems reasonable that the heptamer is enriched under hydrolyzing conditions either in the absence of SNARE substrate or in which a notable fraction of SNARE complexes have been disassembled; the hexamer samples the open state and the seventh protomer enters in the absence of SNARE substrate. Thus, the heptameric state, which is unable to process substrate, may not occur in the cell, as there is likely an abundance of *cis*-SNARE substrate and ATP to capture. Moreover, the XL-MS data cannot disentangle the presence of a hexamer or heptamer because of the small overall changes in residue distances between states. We conclude that NSF/Sec18 is mostly hexameric in the cell unless there is a substantial dearth of substrate to process.

How broadly applicable is this AAA+ side-loading and side-release mechanism beyond NSF/Sec18? The NSF/Sec18 protein family is highly related to the p97/cdc48 family; both are clade 3 AAA+ machines with two AAA+ rings[13]. For p97/cdc48, their substrate is a polyubiquitinated molecule, again imposing topological restraints. In cryo-EM structures of cdc48 with substrate in nonhydrolyzing conditions, the substrate is threaded through the D1 and D2 rings, presumably without any ATP hydrolysis[48]. This study also showed that distal ubiquitin molecules on a substrate do not unfold during cdc48 activity. These observations suggest that this side-loading and side-release mechanism may also be used by cdc48/p97. Furthermore, a split p97 structure was observed when bound to a cofactor[49]. This p97 structure also displayed a coordinated split in its D1 and D2 rings with flattened conformations, similar to those observed for Sec18, suggesting a conserved mechanism.

Lastly, we note that distantly related AAA+ clamp loaders and their clamp also load their nucleic acid substrate from the side[50]. DNA is loaded into an open clamp and AAA+ clamp loader. Once the clamp loader recognizes the substrate, the ATPase is triggered and the clamp loader closes the complex around the DNA. Once the clamp is fully secured, the clamp loader is then released. This shared mechanism suggests the possibility of side loading and release across the entire AAA+ protein family.

## Online content

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

¹Department of Molecular and Cellular Physiology, Stanford University, Stanford, CA, USA. ²Department of Neurology and Neurological Sciences, Stanford University, Stanford, CA, USA. ³Department of Structural Biology, Stanford University, Stanford, CA, USA. ⁴Department of Photon Science, Stanford University, Stanford, CA, USA. ⁵Howard Hughes Medical Institute, Stanford University, Stanford, CA, USA. ⁶Stanford Cryo-Electron Microscopy Center (cEMc), Stanford University, Palo Alto, CA, USA. ⁷Stanford University Mass Spectrometry, Stanford University, Palo Alto, CA, USA. ⁸Department of Biochemistry and Cell Biology, Geisel School of Medicine at Dartmouth, Dartmouth College, Hanover, Germany. ✉e-mail: yousuf@stanford.edu; brunger@stanford.edu

## Methods

### HA–Sec18 *S. cerevisiae* strain preparation and treatment

Briefly, we created the *S. cerevisiae* strain containing HA-tagged Sec18 and treated it with zymolyase and lyticase to dissolve its cell wall but maintain its integrity as a living cell (spheroplast). We then incubated the spheroplasts with DSG, a membrane-permeable protein crosslinking agent, for 30 min before cellular lysis and anti-HA coimmunoprecipitation (co-IP) to isolate Sec18 and any bound proteins to it. This sample was then subjected to SDS–PAGE and the bands corresponding to crosslinked Sec18 were excised and sent for LC–MS/MS analysis. More specifically, tagged Sec18 (pYK175, 3×Flag N-terminal tag, HA C-terminal tag) plasmid was transformed into *S. cerevisiae* S288C using the Frozen-EZ yeast transformation kit (Zymo Research, T2001). Expression of tagged Sec18 was accomplished by growing yeast in 1% raffinose and 1% galactose medium. Cells were spun down and treated with 100 U of zymolyase (Zymo Research, 1004) at 37 °C for 1 h. After incubation, spheroplasts were pelleted with a 1,000*g* spin. DSG diluted in 1× PBS was used to resuspend the pellet and the sample was incubated for 30 min. The excess DSG was then quenched with 1 M Tris and incubated for another 15 min. Cells were washed with cold PBS and were resuspended in lysis buffer (5 mM EDTA, 1 mM ATP, 1 mM TCEP, 150 mM NaCl, 10% glycerol, 0.5% NP40 and 50 mM HEPES, pH 7.4) and glass beads (BioSpec, 11079105). This bead lysis cell solution was vortexed for 5 min at 4 °C three times with ice incubations between each vortexing. The solution was then spun down at 10,000*g* for 5 min at 4 °C. The supernatant was taken and then spun down again at 10,000*g* for 15 min at 4 °C. The supernatant was then taken and subjected to magnetic bead anti-HA co-IP (Thermo Fisher Scientific, 88838). The eluant of the beads was run on an SDS–PAGE gel and the bands corresponding to Sec18 were excised.

### XL-MS sample preparation

Protein samples were embedded in Coomassie-stained gel bands, fixed in 1% acetic acid. The gels were washed several times in 50 mM ammonium bicarbonate, followed by reduction with 10 mM of dithriothreitol (DTT) and incubation for 20 min at room temperature. They were then alkylated using 30 mM acrylamide for 30 min. This was proceeded by overnight digestion at 37 °C with 500 ng of MS-grade trypsin–LysC mix (Promega). After digestion, samples were quenched with formic acid (adjusted to pH ~3) and desalted using MonoSpin C18 solid-phase extraction columns (GL Sciences). Finally, the samples were dried using a SpeedVac (Thermo Fisher Scientific) and exchanged into LC–MS reconstitution buffer (2% acetonitrile with 0.1% formic acid in water) for instrumental analysis.

### LC–MS/MS analysis

MS experiments were performed using an Orbitrap Eclipse Tribrid MS instrument (Thermo Fisher Scientific) attached to an Acquity M-Class ultrahigh-performance LC (UPLC) system (Waters). The UPLC system was set to a flow rate of 300 nl min⁻¹, where mobile phase A was 0.2% formic acid in water and mobile phase B was 0.2% formic acid in acetonitrile. The analytical column was prepared in-house with an inner diameter of 100 μm pulled to a nanospray emitter using a P2000 laser puller (Sutter Instrument). The column was packed with Dr. Maisch 1.9-μm C18 stationary phase to a length of approximately 25 cm. Peptides were directly injected into the column with a gradient of 3–45% mobile phase B, followed by a high-B wash over a total of 80 min. The MS instrument was operated in a data-dependent mode using collision-induced dissociation fragmentation for MS/MS spectra generation.

The raw data were analyzed using Byonic version 5.2.5 (Protein Metrics) to identify peptides and infer proteins. Initial Byonic analyses used a concatenated FASTA file containing the UniProt *S. cerevisiae* proteins and other likely contaminants and impurities. Once sample complexity was determined, a second round of Byonic analyses was completed using a targeted FASTA file, which included only the sequences present

in the samples and allowances for crosslinked peptides with the appropriate linker. Proteolysis with trypsin–LysC was assumed to be fully specific with up to two missed cleavage sites. The precursor ion tolerance was set to 12 ppm. The fragment ion tolerance was set to 0.4 Da. Cysteine modified with propionamide was set as a fixed modification in the search. Variable modifications included oxidation on methionine and acetylation on the protein N terminus. Proteins were held to a false discovery rate (FDR) of 1% using the standard reverse-decoy technique.

The following procedure validated potential crosslinked peptides using Byologic version 5.2.31 (Protein Metrics). Given the number of possible crosslinked peptides observed in these experiments, additional empirical constraints were applied to the potential crosslinked peptides to produce a more rigorous validation set for comparison with other biochemical assays. For the BSG crosslinking studies, crosslinked spectra were required to meet the following criteria: (1) all peptides, crosslinked or native, were filtered to a <1% FDR; (2) a precursor mass error of <7 ppm was required for crosslinked peptides; and (3) the peptide primary sequence was at least 6 aa in length for at least one of the crosslinked peptide pairs. For zero-length crosslinking data analysis, the following additional constraints were added to those described above for BS3 crosslinking: (1) a minimum length of 5 aa was required for both members of the crosslink; (2) an alternative 'XLink' algorithm from Byonic was used to make assignments on the basis of fragmentation of both peptides rather than just crosslink partner mass; (3) the crosslinks were assumed valid only if the protein contained the lysine crosslink partner; and (4) at least two crosslinked peptide spectra were assigned to the linkage. Following this, identified crosslinked peptides were further categorized into three groups: high confidence, medium confidence and low confidence, where 'high confidence' succeeded on all these rules and 'low confidence' failed at least three of these rules.

### Preparation of y20S in the nonhydrolyzing condition

The y20S complex consists of the ySNARE complex, the Sec18 oligomeric complex and Sec17 adaptor proteins and it was assembled in a multistage workflow. Starting with the yeast exocytic complex, individual SNARE components Sec9 (401–651), Sso1 (1–265) and Snc1 (1–93) were purified individually. All three proteins were cloned into the pET28b plasmid vector with a 6×His tag and a tobacco etch virus (TEV) cleavage sequence N-terminal to the SNARE protein sequence.

For Sec9, we used a lysis buffer of 50 mM NaPi, 300 mM NaCl, 10 mM imidazole, 0.5 mM TCEP, 1% Triton X-100 and SIGMAFAST protease inhibitor cocktail tablet (pH 7.5; Sigma-Aldrich, S8830), which was used to resuspend bacterial cell pellets from an 8-L culture of autoinduced one-shot BL21(DE3) *Escherichia coli* (Thermo Fisher Scientific, C600003). This resuspended lysate was subjected to sonication for 20 min (3 s on, 9 s off, 60% amplitude), followed by a 30-min 4,000*g* spin and a 1-h 40,000*g* spin. The supernatant was bound (1 h, 4 °C), run through a Ni-NTA column and washed with 50 mM NaPi, 30 mM Imidazole, 300 mM NaCl, and 0.5 mM TCEP buffer (pH 7.5). The sample was then eluted with 50 mM NaPi, 400 mM Imidazole, 300 mM NaCl and 0.5 mM TCEP buffer (pH 8). This sample was then digested overnight with TEV protease in a dialysis buffer (20 mM Tris, 250 mM NaCl, 0.5 mM TCEP and 1 mM EDTA, pH 8). The cleaved sample was then subjected to a MonoS 5/50 GL with buffer A consisting of 20 mM Tris, 50 mM NaCl, 0.5 mM TCEP and 1 mM EDTA (pH 8) and buffer B consisting of 20 mM Tris, 500 mM NaCl, 0.5 mM TCEP and 1 mM EDTA (pH 8). Taking the majority peak corresponding to Sec9, we then subjected this peak to size-exclusion chromatography (SEC) with the HiLoad 16/60 Superdex 200. The peak corresponding to Sec9 was concentrated and flash-frozen. We followed the same protocol for individually purifying Snc1 and Sso1.

Once all three individual SNAREs were purified, a 1:1:1 molar ratio of these proteins was added to a 6 M GdHCl solution. This solution was then slowly dialyzed overnight at 4 °C into a solution of 250 mM NaCl and 50 mM HEPES (pH 7.6). This sample was then diluted to a NaCl concentration of 75 mM and then subjected to MonoQ with the

low-salt buffer at 75 mM and the high-salt buffer at 500 mM with both buffers in 50 mM HEPES pH 7.6. The peak corresponding to a 1:1:1 ratio of all three bands, indicative of the yeast exocytic complex, was then concentrated for y20S formation without freezing.

The complete Sec18 protein coding sequence was cloned into the pMZ0002/pYK103 backbone with a 6×His tag and TEV cleavage sequence N-terminal to Sec18. The 8-L culture of one-shot BL21(DE3) *E. coli* (Thermo Fisher Scientific, C600003) was spun down and resuspended in lysis buffer of 100 mM HEPES, 500 mM KCl, 5 mM ATP, 5 mM $MgCl_2$, protease inhibitor tablets and benzonase (pH 7.5). This lysate was sonicated for 20 min (3 s on, 9 s off, 60% amplitude) and clarified with a 45,000*g* spin for 30 min. The lysate was supplemented with 20 mM Imidazole bound to Ni-NTA beads (4 °C) and washed with 20 mM HEPES, 480 mM KCl, 0.5 mM ATP, 0.5 mM TCEP, 20 mM imidazole, 1 mM $MgCl_2$ and 10% glycerol (pH 7.5). The sample was then eluted with 20 mM HEPES, 480 KCl, 0.5 mM ATP, 0.5 mM TCEP, 500 mM imidazole, 1 mM $MgCl_2$ and 10% glycerol (pH 7.5). The protein-containing peak was then equilibrated in SEC buffer 20 mM PIPES, 125 mM KCl, 0.2 M sorbitol, 5 mM $MgCl_2$, 2 mM ATP, 2 mM DTT and 10% glycerol (pH 6.8) and injected a HiLoad 16/60 Superdex 200. The fractions were pooled, concentrated and snap-frozen for later use. The band right below Sec18 is an oligomeric *E. coli* contaminant, which is removed in the final steps of y20S complex purification (Supplementary Fig. 2a,c). Moreover, this contaminant is absent in the protocol for preparing high-purity Sec18 (Supplementary Fig. 6b).

The full Sec17-coding sequence was cloned with a 10×His tag and a TEV cleavage sequence fused N-terminally. The 8-L culture of one-shot BL21(DE3) *E. coli* was spun down and resuspended in a lysis buffer of 50 mM NaPi, 300 mM NaCl, 20 mM imidazole and 0.5 mM TCEP (pH 8). This mixture was then sonicated (3 s on, 9 s off, 60% amplitude) and clarified at 185,000*g* for 30 min. This supernatant was then incubated with Ni-NTA beads for 1 h and then washed with lysis buffer. The protein was eluted with lysis buffer supplemented with 500 mM imidazole and 5 mM EDTA. TEV protease was added to pooled fractions and slowly dialyzed overnight at 4 °C against 50 mM Tris, 100 mM NaCl, 1 mM EDTA and 0.5 mM TCEP buffer (pH 8). This sample was then diluted to a NaCl concentration of 50 mM and run on a MonoQ 10/100 along a salt gradient of 50 mM to 500 mM. The major peak corresponding to Sec17 was then pooled and injected into a HiLoad 16/60 Superdex 200 SEC. The major peak from this run was taken, pooled and snap-frozen for later use.

To assemble the entire y20S complex, Sec17 was added to the freshly prepared yeast exocytic SNARE complex, followed by EDTA-quenched Sec18 with a final ratio of Sec18:SNARE:Sec17 at 1:1.67:10, where a total of 5,580 pmol of Sec18 was used, in 50 mM Tris, 150 mM NaCl, 1 mM TCEP, 1 mM ATP and 1 mM EDTA (pH 8). Following this assembly, this mixture was injected onto a Superose 6 10/300 increase column and the peak corresponding to the y20S complex, as assessed by elution volume and corresponding SDS–PAGE gel, was concentrated to ~40 mg ml⁻¹.

### Single-particle cryo-EM grid preparation of y20S in the nonhydrolyzing condition
Quantifoil R1.2/1.3 200-mesh gold grids were treated with chloroform and dried overnight. Grids were glow-discharged and 5 µl of the sample (at a final concentration of 20 mg ml⁻¹ with 0.05% v/v Nonidet P-40) was blotted onto the grids and further blotted and vitrified in liquid ethane using an FEI Vitrobot (Thermo Fisher Scientific). Grids were blotted for 4 s with a 5-s wait time (reduced to 3 s in hydrolyzing condition).

### Sample preparation and single-particle cryo-EM of y20S in the hydrolyzing condition
Samples were prepared identically to y20S in the nonhydrolyzing condition, except that excess $MgCl_2$ was blotted onto grids immediately before vitrification and the total blot and wait time was reduced to 7 s before freezing.

### Sample preparation and single-particle cryo-EM of substrate-free Sec18 in the hydrolyzing condition
An alternative protocol was required to prepare high-purity, substrate-free Sec18 because low-purity Sec18, which can form y20S, was not amenable to high-quality cryo-EM studies. Therefore, we used a different protocol for preparing NSF[10] except that, at the final reassembly step, we used $MgCl_2$ instead of EDTA. The cryo-EM studies were performed similarly to the studies of Y20S in the nonhydrolyzing condition.

### Sample preparation and single-particle cryo-EM of NSF in the hydrolyzing condition
A sample of high-purity NSF was prepared as described above. This sample was exchanged into a buffer supplemented with 1 mM $MgCl_2$ instead of EDTA and incubated before a final SEC and freezing.

### Single-particle cryo-EM data collection, processing and model building
The single-particle cryo-EM data collection and processing workflow are described thoroughly in Extended Data Table 1 and Supplementary Fig. 2. An FEI Titan Krios (Thermo Fisher Scientific) cryo-EM instrument with either a K3 Gatan or a Falcon F4i camera was used for data collection. Micrographs were analyzed using a combination of initial RELION 3.1 (ref. 52) processing followed by additional rounds of 3D classification or heterogenous refinement and final refinements in cryoSPARC version 4 (ref. 53). A general workflow description is provided in Supplementary Fig. 2 and specific dataset information is provided in Supplementary Figs. 2–6.

y20S models were constructed by first docking the crystal structure of the exocytic SNARE complex (Protein Data Bank (PDB) 3B5N) and PDF/AlphaFold2 (ref. 54) models for Sec17 (1QQE/AF-P32602-F1-v4) into the unsharpened density first obtained in RELION 3.1 (ref. 52). Next, sharpened cryoSPARC version 4 (ref. 53) maps were used to model the Sec18 complex and the region of Sso1 that extended from the spire into the D1 pore. For class 1, 3DFlex analysis in cryoSPARC version 4 was performed. A mixture of the RELION 3.1 and cryoSPARC version 4 unsharpened and sharpened maps and the 3Dflex map were used to model the domain outside of the D2 ring. The resulting atomic models were iteratively refined with a combination of Coot[55], PHENIX real-space refinement[56] and ISOLDE[57].

Substrate-free Sec18 and NSF models were built de novo in Coot using cryoSPARC sharpened maps and iteratively refined with ISOLDE and PHENIX. AlphaFold2 model of Sec18 was used as a starting reference for model building (AF-P18759-F1-v4).

### Structural and conformational analysis
Statistical structural analysis was performed to identify the most notable backbone motions between the different structural states of Sec18 without substrate, as observed in Fig. 6. Using Python 3, principal component analysis (PCA) on backbone $\phi$ and $\psi$ angles was performed and the single mode that contributed most to the variance (>70%) was analyzed. y20S models were then clustered in this reduced five-dimensional space. When comparing the substrate-free and substrate-engaged states, ensembles consistent with a given map were generated to avoid model-building bias using PHENIX's simple molecular dynamics implementation for 2,000 steps followed by a round of phenix.refine[56]. These ensembles were then subjected to PCA of backbone $\phi$ and $\psi$ angles and the mode that contributed to the largest variance between the two clusters for the two classes was used to determine which residues contributed most to the difference in structures. More details can be found in the conformational tool package on GitHub with annotated and commented code (https://github.com/YousufAKhan/ConformationalAnalysis/).

### smFRET disassembly and reassembly assay
smFRET assays with linked neuronal SNAREs, NSF, α-SNAP and reagents were prepared as previously reported[11]. Briefly, linked SNARE complex

(L-SNARE) composed of SNAP25A, synaptobrevin 2 and syntaxin 1A was biotinylated and flowed onto the chamber that was passivated by 10 mg ml⁻¹ egg phosphatidylcholine 50-nm liposomes coated with 1 mg ml⁻¹ biotinylated BSA to mimic the lipid environment inside the cell. Then, 0.1 mg ml⁻¹ streptavidin was added to surface-tether the biotinylated and labeled L-SNARE complexes. L-20S was then assembled by first adding αSNAP and then later NSF (if in disassembly conditions). L-SNARE was diluted such that about 500 molecules of L-SNARE per $45 \times 90$-$\mu m^2$ field of view were visible. The labels for the Habc domain were at residue positions 35 and 105 and labels for the L-SNARE complex were at residue position 249 of syntaxin and residue position 82 of synaptobrevin, identical to a previous study[11]. The Habc label positions were at the tips of the α-helices positioned away from the D2 pore and the Sec18 complex (Fig. 5a). Thus, one would expect that 20S complex formation does not affect the quantum yield and conformational dynamics of the dyes, which is indeed what we observed. The labeling was performed by substituting the residues at these positions to cysteines and stochastic conjugation with Alexa 555 and Alexa 647 dyes.

All smFRET experiments were performed on a prism-type total internal reflection fluorescence microscope using 532-nm (green) laser (CrystaLaser) excitation. Two observation channels were created by a 640-nm single-edge dichroic beamsplitter (FF640-FDi01 25 × 36, Shemrock); one channel was used for the fluorescence emission intensity of the Alexa 555 dye and the other channel was used for the Alexa 647 dye. The two channels were recorded on two adjacent rectangular areas ($45 \times 90$ $\mu m^2$) of a charge-coupled device camera (iXon+ DV 897E, Andor Technology). The imaging data were recorded with the smCamera program (T. Ha, Johns Hopkins University)[58]. Flow chambers were assembled by creating a 'sandwich' consisting of a quartz slide and a glass coverslip that were both coated with polyethylene glycol (PEG) molecules consisting of 0.1% (w/v) biotinylated PEG except when stated otherwise and using double-sided tape to create up to five flow chambers.

Fluorescence intensity time traces were recorded by the smCamera[58] program; peaks were selected and correlated in the two channels as previously described[59]. The time traces were imported into tMaven[51] for smFRET analysis. The standard recommended tMaven workflow was used. In short, all traces were evaluated manually and not subject to any automated trace preprocessing in tMaven. Traces that did not have single photobleaching events were not considered. After obtaining all traces that fit these criteria, the automatic photobleaching algorithm was used to remove regions that were photobleached for analysis of the FRET states. Following this, the vbGMM model was used to model the transition between states with an assumption of two states (that is, high and low). Following this, the tMaven 'one-dimension histogram' tool was used to generate histograms of the distribution of states of the raw FRET efficiency $E = I_A/(I_A + I_D)$, where $I_A$ and $I_D$ are the acceptor and donor fluorescence intensities, respectively; no corrections for bleed through, detection efficiency and quantum yield were applied.

#### Crosslinking assays
Cysteines were introduced into Sec9 and Sso1 for labeling with Oregon green maleimide 488 (O6034). UAG stop codons were introduced at V57 and F83 in Sso1 for crosslinking. Cys–Sec9, Cys–Sso1 (with or without stop codon depending on if used for crosslinked) and 6×His–TEV–Snc1 were coexpressed (if crosslinked, also transformed with pEVOL-pAzF (Addgene, 31186), 2 mM AzF and 0.2% v/v arabinose) and induced with IPTG. Cell pellets were lysed with 50 mM Tris, 200 mM NaCl, 1% v/v Triton X-100, 20 mM imidazole, 0.5 mM TCEP and protease inhibitor tablets (pH 7). Lysate was sonicated for 20 min (3 s on, 9 s off, 60% amplitude) and then clarified at 40,000$g$ for 30 min. Following a 1-h incubation at 4 °C, the beads were washed with two different buffers. Buffer 1 was simply a lysis buffer without Triton X-100 and buffer 2 was a high-salt wash of 50 mM Tris, 1 M NaCl, 50 mM imidazole and 0.5 mM TCEP (pH 7). The complex was eluted with 50 mM Tris, 200 mM

NaCl, 350 mM imidazole and 0.5 mM TCEP (pH 7). The sample was then digested overnight with TEV protease and dialyzed into 20 mM Tris, 200 mM NaCl and 0.5 mM TCEP (pH 7). This sample was then injected onto a HiLoad 16/60 Superdex 200. The fractions containing all three in equimolar ratios were taken and concentrated. Samples were labeled with Oregon green maleimide 488 in molar excess overnight before cleanup with a Zeba spin column (Thermo Fisher Scientific, 89889). If the sample was meant to be crosslinked, it was subjected to ultraviolet (UV) irradiation before Oregon green maleimide labeling. The buffers used for these crosslinking assays were the same as the final dialysis buffer but were degassed before UV irradiation and Oregon green maleimide labeling.

For native gel disassembly assays, reactions were assembled in test tubes, initiated with MgCl₂ and quenched with EDTA. Samples were then loaded into Any kD Mini-Protean gels (Bio-Rad, 4569033) and used Tris–glycine running buffer (Bio-Rad) supplemented with 1 mM ATP to prevent Sec18 from falling apart in the gel. Gels were first visualized in an iBright 1500 (Thermo Fisher Scientific) to visualize ySNARE bands only before Coomassie staining. Gel densitometry was assessed in ImageJ.

#### FRET and fluorescent dequenching disassembly assays
Neuronal SNARE complex disassembly assays were carried out as previously reported[10], leveraging fluorescence dequenching of the neuronal SNARE complex labeled with Oregon green maleimide 488. For yeast SNARE complex disassembly assays, a strategy was devised to tag Snc1 with mTurq and Sso1 with mVenus. When assembled, excitation by a 434-nm laser led to mVenus emission and 434-nm laser excitation led to mTurq emission. Disassembly was followed by monitoring excitation and emission at 434 and 474 nm such that, as the SNARE complex was disassembled, mTurq emission increased. We used the same instrument (FlexStation II 384, Molecular Devices) and a 384-well format to perform both the dequenching and the FRET assays that were initiated by titrating MgCl₂ in excess of the reactions.

#### Vacuolar fusion assay
Assays were performed as previously reported[40,41] with the same substrate-free Sec18 protein sample batch that was purified and used for Cryo-EM. Briefly, proteoliposomes with vacuolar SNAREs were prepared with either phycoerythirin–biotin (PhycoE) or Cy5 and incubated in a fluorescence plate reader for 40 min. FRET between PhycoE and Cy5 was measured at 1-min intervals. Reaction mixtures contained HOPS complex, Sec17 and variable amounts of Sec18.

#### Reporting summary
Further information on research design is available in the Nature Portfolio Reporting Summary linked to this article.

### Data availability
The MS proteomics data were deposited to the ProteomeXchange Consortium through the PRIDE partner repository with the dataset identifier PXD062002. All cryo-EM maps and coordinates were deposited to the EM Data Bank and PDB, respectively, under the following accession codes: Y20S nonhydrolyzing condition, class 1, EMD-45883 and PDB 9CRU; class 2, EMD-48826 and PDB 9N22; class 3, EMD-45885 and PDB 9CRX; class 4, EMD-49380 and PDB 9NG2; class 5, EMD-49522 and PDB 9NLU; class 6, EMD-49524 and PDB 9NLW; class 7, EMD-49526 and PDB 9NLY; class 8, EMD-49527 and PDB 9NLZ; class 9 (all merged, focusing on D1–D2), EMD-49528 and PDB 9NM1; Y20S hydrolyzing condition, class 1, EMD-70472 (no model created); class 2, EMD-49801 and PDB 9NUD; class 3, EMD-49802 and PDB 9NUE; class 4, EMD-49824 and PDB 9NUZ; Sec18 hydrolyzing condition, class 1, EMD-49826 and PDB 9NV1; class 2, EMD-49825 and PDB 9NV0; NSF hydrolyzing condition, class 1, EMD-49831 and PDB 9NV9; class 2, EMD-49833 and PDB 9NVD. The smFRET data were deposited to the Stanford Data Repository

(https://purl.stanford.edu/bm031pm6709). All data are available with the manuscript online and materials can be obtained from Addgene. Source data are provided with this paper.

## Code availability

The conformational tool package was deposited to GitHub (https://github.com/YousufAKhan/ConformationalAnalysis) and smFRET related scripts can also be found on GitHub (https://github.com/brungerlab/single_molecule_matlab_scripts).

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

## Acknowledgements

We thank the late W. I. Weis and J. D. Puglisi for stimulating discussions. We thank the National Institutes of Mental Health for support (RO1MH63105 to A.T.B. and 1F31MH134477 to Y.A.K.). Y.A.K. was additionally supported in part by a National Science Foundation Graduate Research Fellowships Program and the Knight-Hennessy Scholarship. K.I.W. was supported by a postdoctoral fellowship from the Helen Hay Whitney Foundation, supported by the Howard Hughes Medical Institute (HHMI). W.T.W. is supported by a grant from the National Institutes of Health (NIH; 2R35GM118037). We thank the Vincent Coates Foundation MS Laboratory, Stanford University MS (RRID: SCR_017801) for use of the Thermo Orbitrap Eclipse nanoLC/MS system (RRID: SCR_022212) that was purchased with funding from the NIH Shared Instrumentation Grant 1S10OD030473, the Stanford Cancer Institute Proteomics/MS Shared Resource (NIH P30 CA124435). This article is subject to HHMI's open access to publications policy. HHMI laboratory heads have previously granted a nonexclusive CC BY 4.0 license to the public and a sublicensable license to HHMI in their research articles. Pursuant to those licenses, the author-accepted manuscript of this article can be made freely available under a CC BY 4.0 license immediately upon publication.

## Author contributions

Y.A.K. and A.T.B. conceptualized and designed the experiments. K.I.W., B.S. and E.M. assisted Y.A.K. with the cryo-EM data collection. Y.A.K. performed and analyzed all the experimental data. K.I.W. assisted Y.A.K. with the cryo-EM data processing and model building. R.A.F. and L.E. assisted with the protein purification. G.M., F.L. and T.M. assisted with the high-resolution MS proteomics data collection and analysis. K.D. assisted with the bulk disassembly assays. U.B.C. collected and assisted with the smFRET data. W.T.W. performed and analyzed the vacuolar fusion assays. Y.A.K., A.T.B. and K.I.W. wrote the manuscript.

## Competing interests

The authors declare no competing interests.

## Additional information

**Extended data** is available for this paper at https://doi.org/10.1038/s41594-025-01590-w.

**Correspondence and requests for materials** should be addressed to Yousuf A. Khan or Axel T. Brunger.

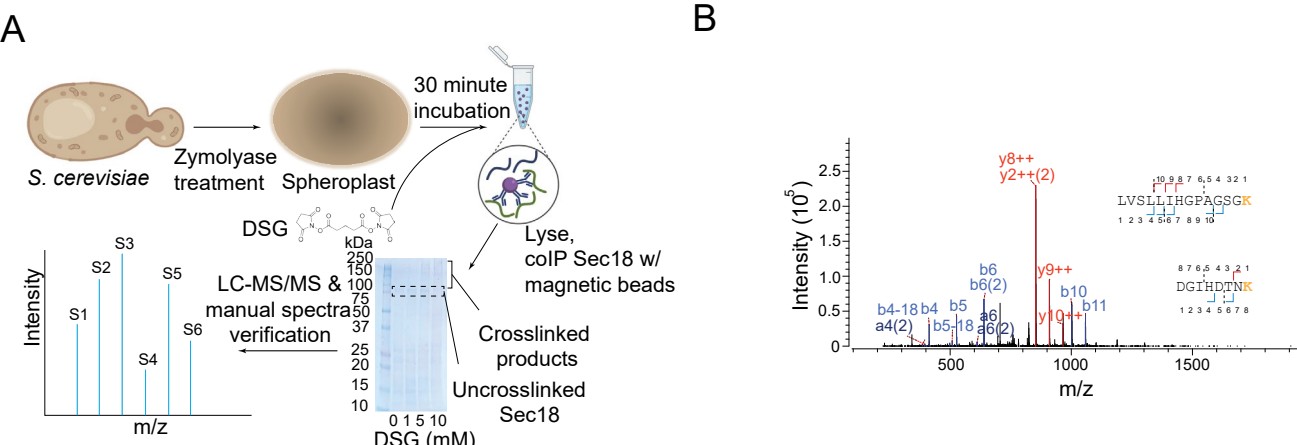

**Extended Data Fig. 1 | In vivo mass-spec workflow and additional analysis. a**, workflow of how samples were processed before mass-spec. **b**, Representative mass-spectrometry raw data of high confidence (Methods), suggesting that the Habc domain is crosslinked to the D2 domain. In the sequence diagram, the cross-linked lysine residues are shown in orange.

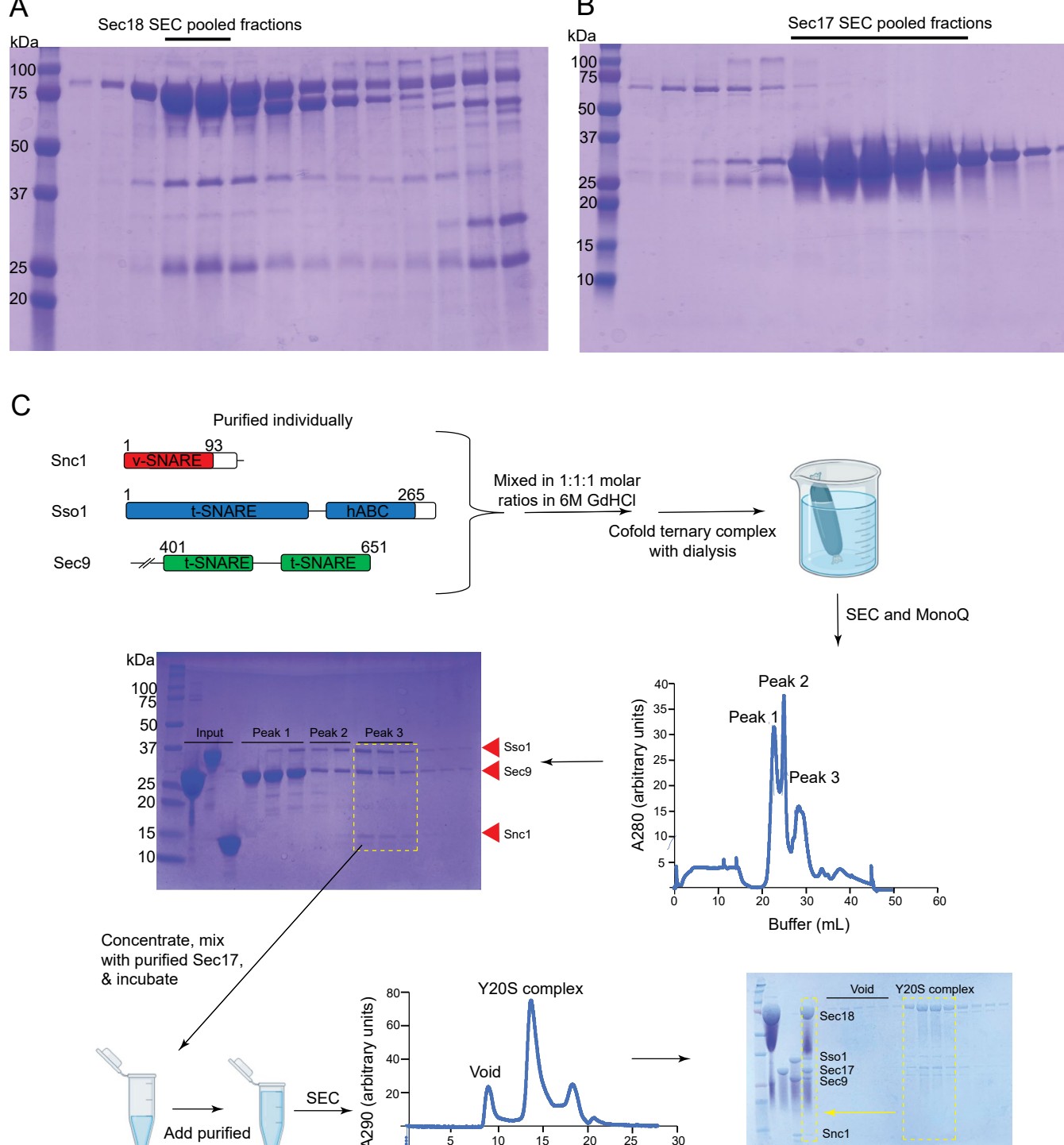

**Extended Data Fig. 2 | Purification and assembly of y20S complex. a**, Sec18 SDS-PAGE gel after the final SEC run. Pooled fractions were frozen and stored. **b**, Sec17 SDS-PAGE gel after final SEC run. Pooled fractions were frozen and stored. **c**, Schematic of y20S assembly. Snc1, Sso1, and Sec9 were purified individually before being co-folded for complex formation (Methods). The resulting SNARE complex was added to purified Sec17, EDTA was added, and then purified Sec18 was added. The resulting y20S complex was purified by SEC, and fractions were pooled. The final gel shows reference inputs for Sec18, Sec17, and yeast SNARE complex, the final concentrated y20S complex, and the fractions from SEC that were pooled (yellow lines).

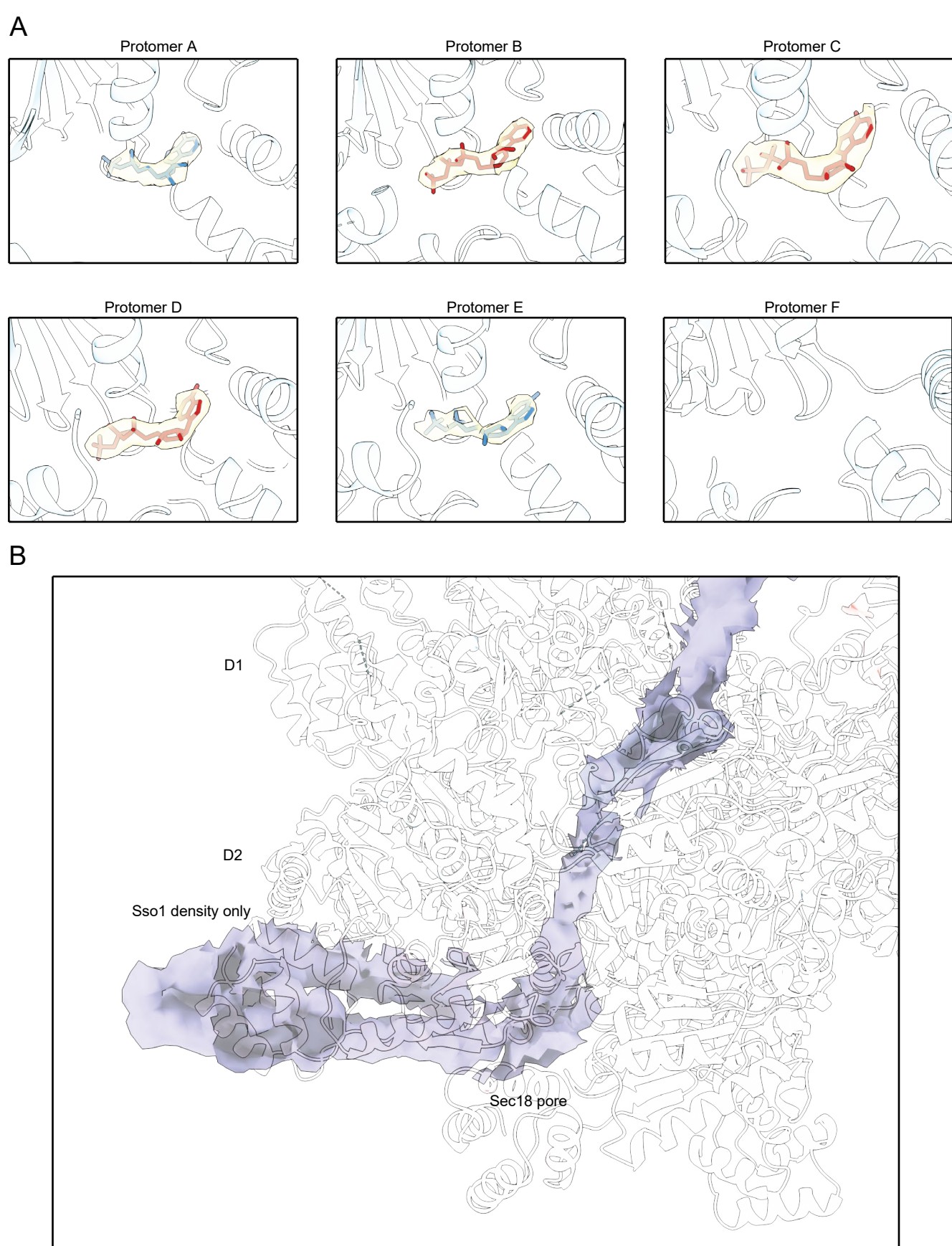

**Extended Data Fig. 3 | Raw cryo-EM densities. a**, Nucleotide densities of Y20S EDTA D1 and D2 rings. The cryo-EM map is depicted as yellow, ATP as red, and ADP as blue. **b**, Close-up view of a composite cryo-EM map (light purple) around Sso1 in the D2 ring. The D1 and D2 rings are depicted as transparent cartoons.

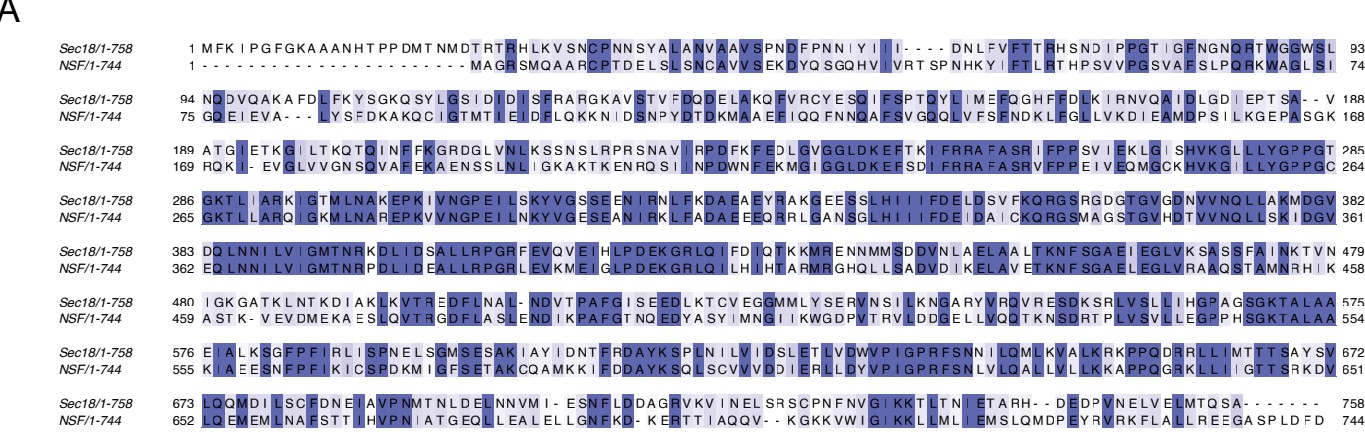

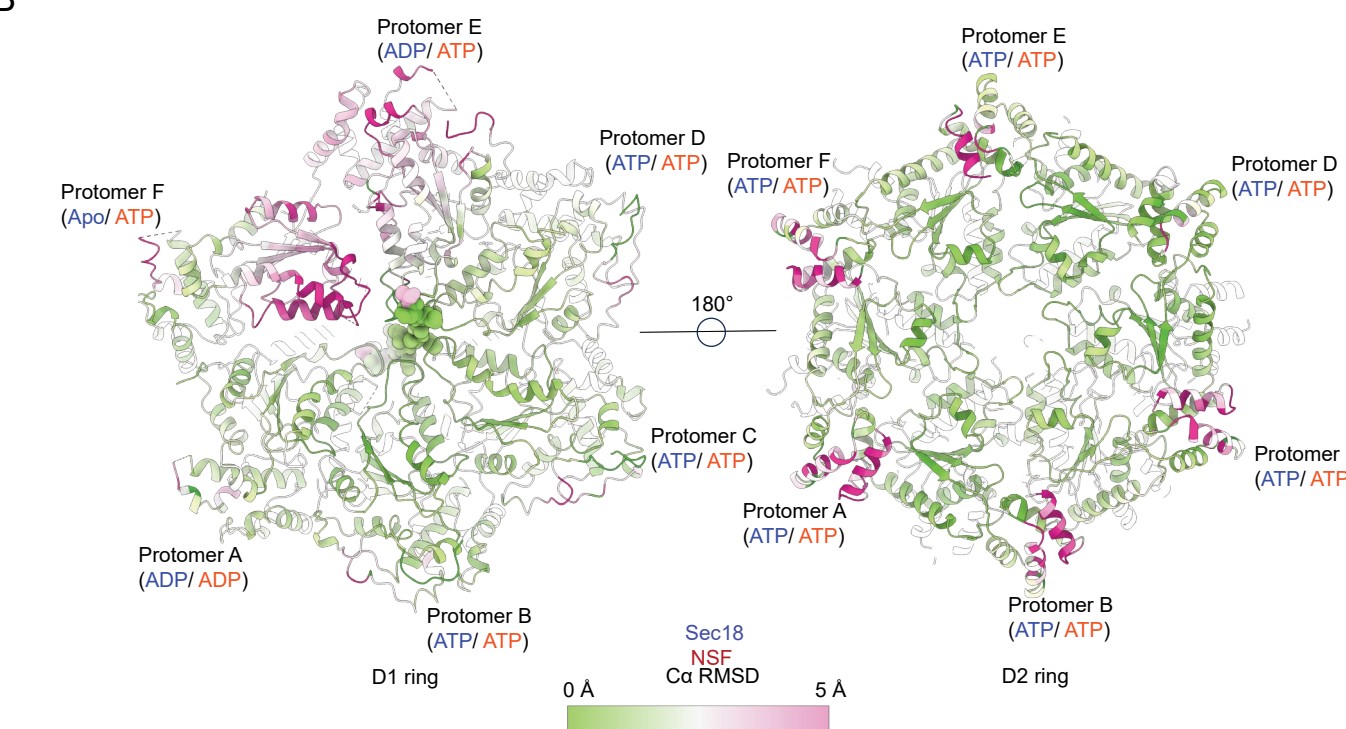

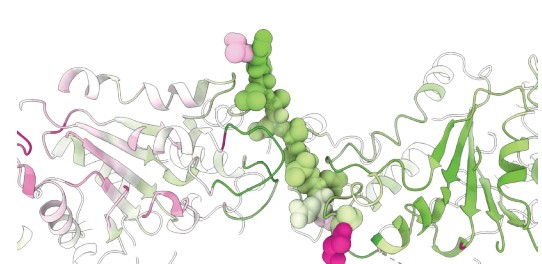

**Extended Data Fig. 4 | Sequence and structural conservation between NSF and Sec18. a**, Primary sequence alignment between NSF *c. griseus* and Sec18 *s. cerevisiae*. **b**, cartoon representation of an atomic model of Sec18 D1/D2 focused model colored by Cα root-mean-square-difference (RMSD) to the structure of the binary SNARE complex with NSF and α-SNAP. **c**, Focused view of substrate in D1. The substrate bound to the D1 pore is represented as spheres.

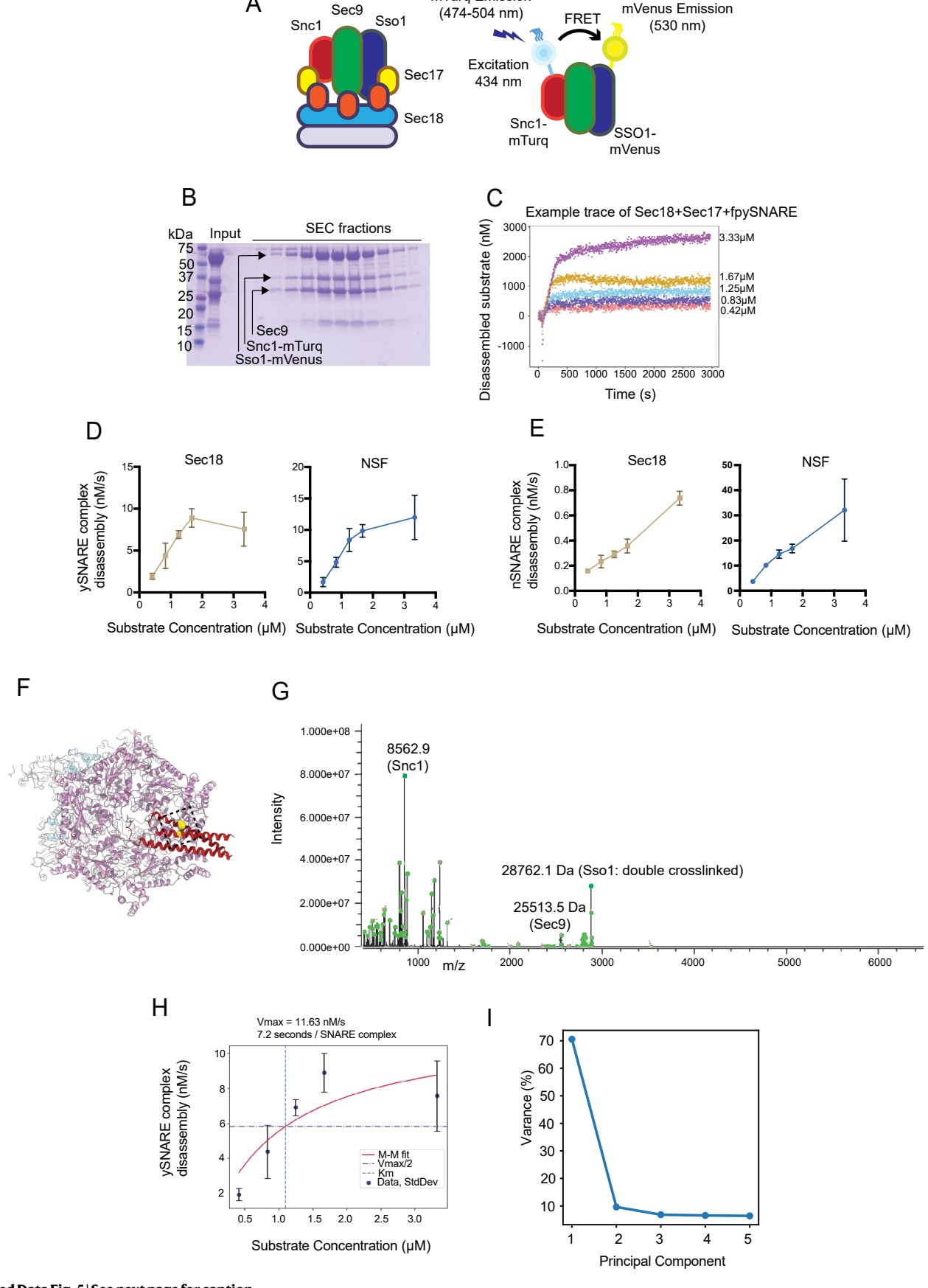

**Extended Data Fig. 5 | See next page for caption.**

**Extended Data Fig. 5 | Cross-complementation disassembly assays. a**, Scheme of fluorescent-protein labeling. Snc1 was labeled with mTurq, and SSO1 was labeled with mVenus (Methods, referred to as fpySNARE). **b**, SDS Page gel of the final SEC purification of y20S complex. **c**, Representative traces from the disassembly assay (Methods). The mTurq emission is monitored, and an increase in emission as the complex suggests that SNARE disassembly occurs (loss of fluorescence resonance energy transfer to mVenus upon disassembly). The concentration of the *cis*-SNARE complex is increased in each sample trace.

**d-e**, Disassembly plots of SNAREs and NSF/Sec18 combinations (n = 3). **f**, Residues that were replaced with p-azido-l-phenylalanine are shown as yellow spheres in a black box. **g**, MS raw data of double crosslinked yeast exocytic SNARE complex. **h**, Michaelis-Menten diagram of the y20S disassembly reaction (n = 3). The red line shows the fitted line used to calculate M-M metrics, whereas the black points and error bars show the actual data. **i**, Variance contributed by each principal component. All error bars represent SEM and center points represent mean for **d-e**, and **h**.

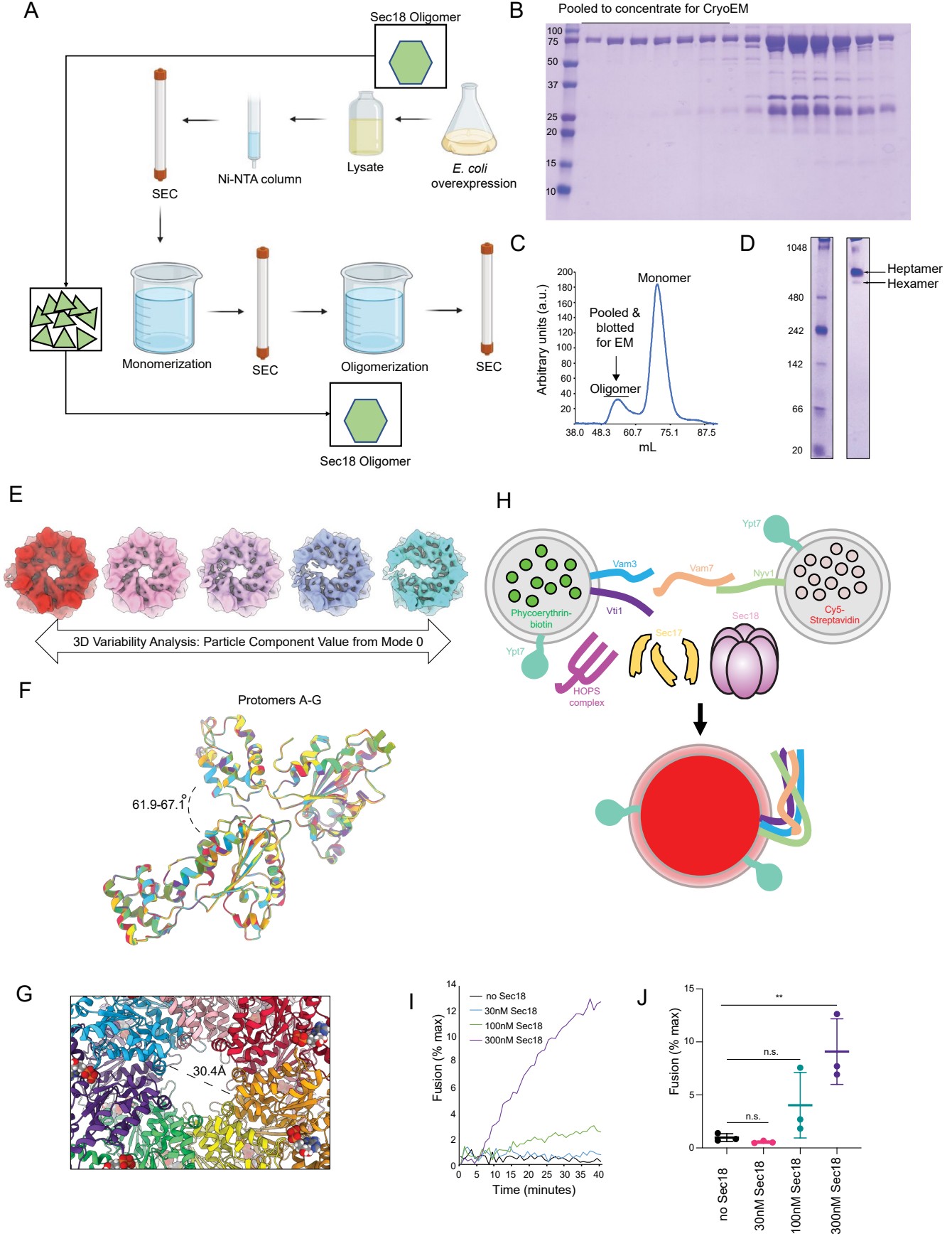

**Extended Data Fig. 6 | See next page for caption.**

**Extended Data Fig. 6 | Functional studies of Sec18. a**, Schematic of purification of Sec18. **b**, SDS-PAGE gel of final SEC fractions and the fractions that were pooled for Cryo-EM studies of the Sec18 complex and the vacuolar fusion assay. **c**, Corresponding SEC trace showing the Sec18 oligomer eluting at expected volume. **d**, Native gel of final sample showing two oligomeric species. **e**, Representative slices of Sec18 complex under the hydrolyzing condition, subjected to 3DVA revealing split hexamer class. **f**, Alignment of all seven protomers, where each protomer is colored uniquely. **g**, Cartoon depicting the large pore in the heptameric state of Sec18 complex under the hydrolyzing condition. **h**, Cartoon of the vacuolar fusion assay (Methods) used to determine the function of the same Sec18 sample used for Cryo-EM studies of the y20S complex. Fusion only occurs when Sec18 prepares SNAREs for fusion, leading to FRET between the entrapped proteins from each fusion partner. **i**, Representative traces of fusion activity. **j**, Dot-plots of all activity assays, error bars denote SEM and line denotes mean (n = 3). A one-way ANOVA was performed, comparing all groups to the no Sec18 condition. ** represents a p-value less than 0.01.

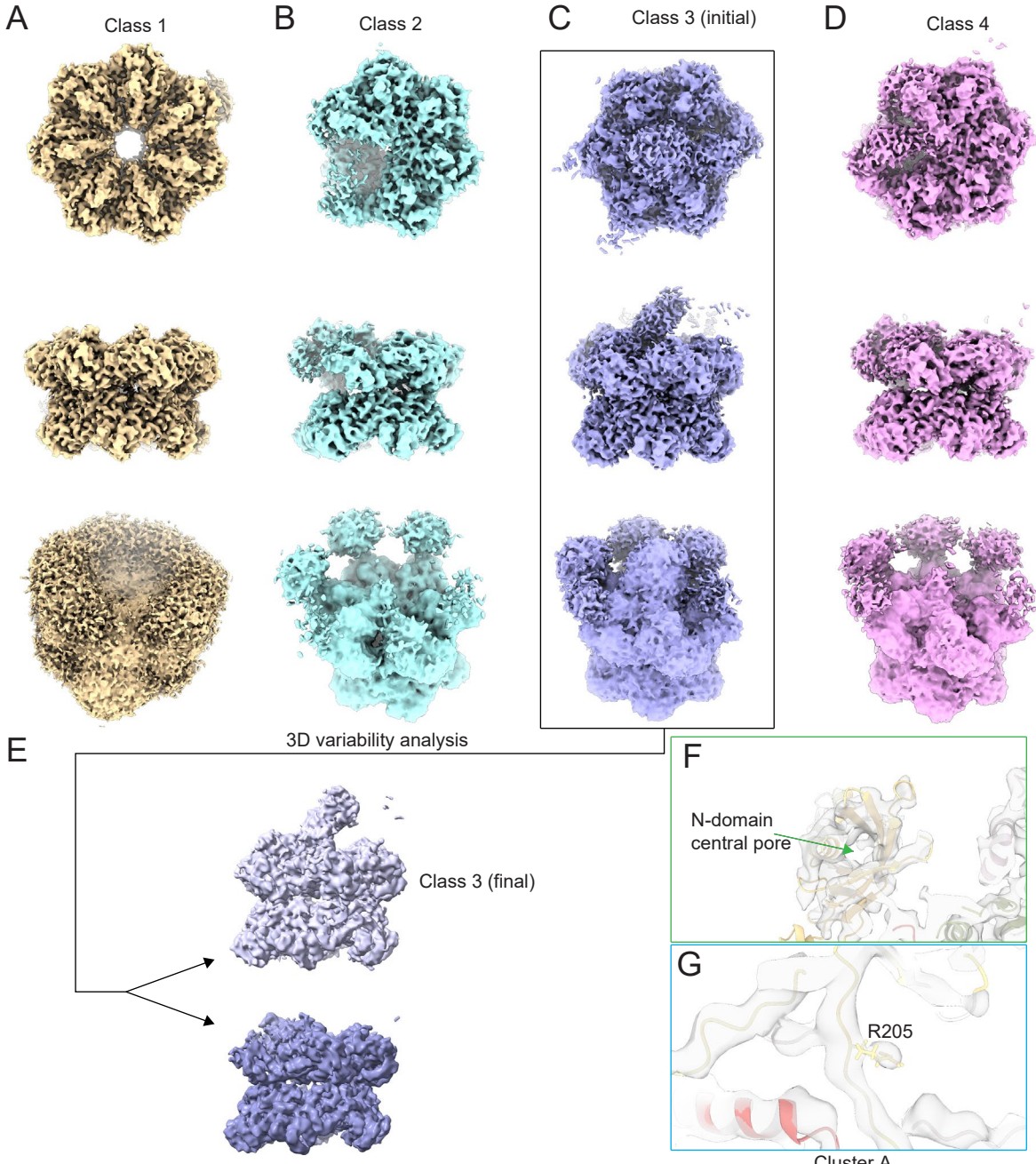

**Extended Data Fig. 7 | Cryo-EM structures of substrate-free NSF in the hydrolyzing condition. a-d**, Cryo-EM maps (sharpened cryoSPARC) in three orientations (top to bottom rows) of the four classes of NSF in the hydrolyzing condition. The NSF heptamer (class 1) is designated as class 1. Classes 2-4 showed hexameric NSF in similar orientations with varying resolution for the substrate, so the best one, class 3 (initial) (c), was chosen for 3DVA analysis in CryoSPARC.

**e**, The main principal component of the 3DVA analysis was used to select a subset of particles clearly showing an NSF N-terminal domain that is engaged with the D2 ring and then re-refined against this subset of particles to yield the final map for class 3. **f-g**, insets showing that an N-domain acts as a bound substrate in class 3 (final).

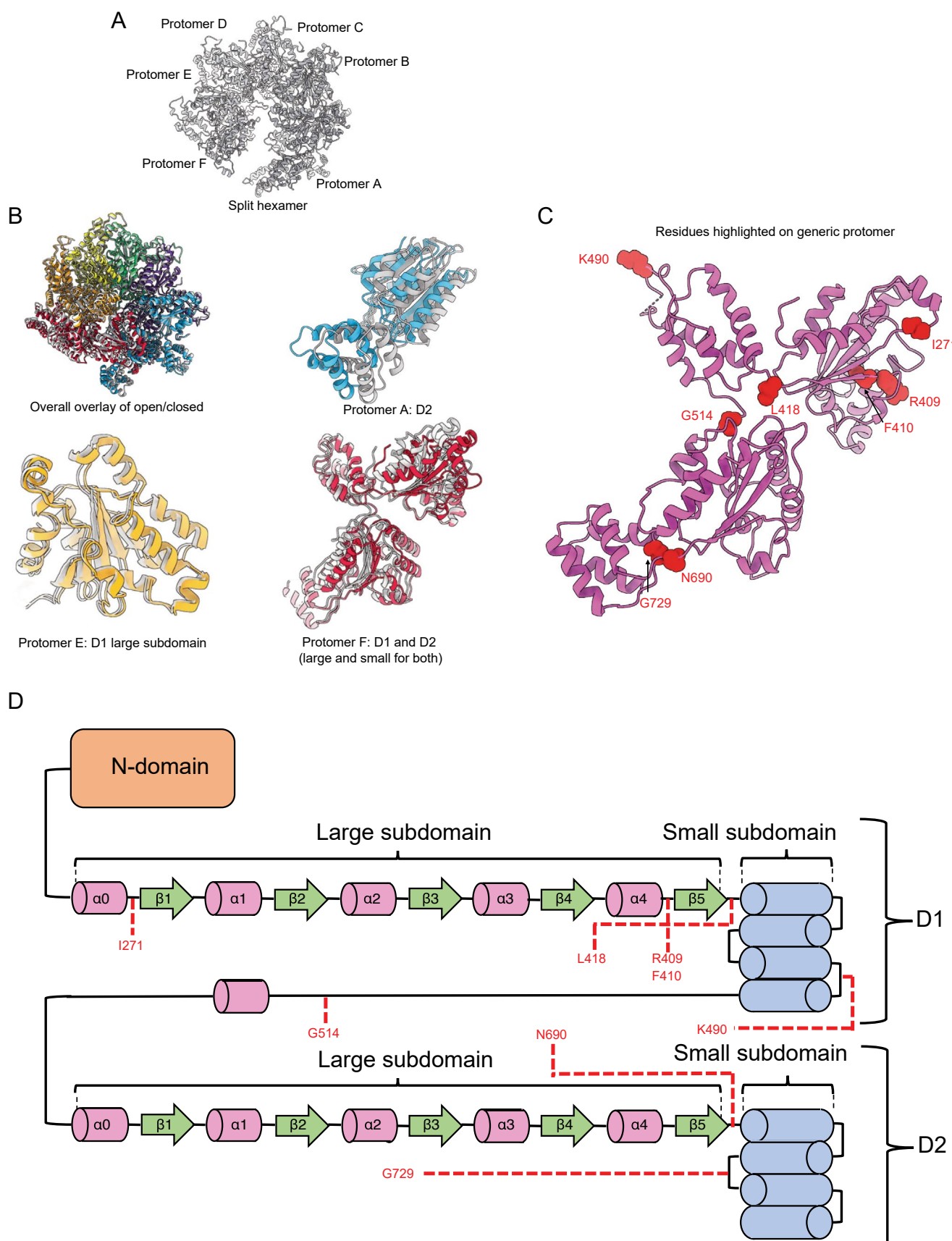

Extended Data Fig. 8 | See next page for caption.

**Extended Data Fig. 8 | Conformational analysis of coordinated D1, D2 ring opening. a**, Model of split Sec18 hexamer corresponding to Fig. 6. **b**, Top left panel: superimposition of substrate-free Sec18 closed (grey) and open (colored) corresponding to Fig. 6. Top right panel: Superimposition of D2 protomer A. Bottom left panel: Superimposition of protomer E D1, large subdomain. Bottom right panel: Superimposition of entire protomer F (no N-domain). **c**, Residues above 20% variance contribution are shown as red atoms on a generic protomer. **d**, Domain layout of a protomer with red labels indicating the identified residues that differ most significantly between closed and open Sec18 D1 and D2 rings.

## Extended Data Table 1 | CryoEM refinement and validation information

| | Y20S, non-hydrolyzing condition | | | | | | | | | Y20S, hydrolyzing condition | | | | Sec18, hydrolyzing condition | | NSF, hydrolyzing condition | |
|---|---|---|---|---|---|---|---|---|---|---|---|---|---|---|---|---|---|
| | #1 Class 1 (EMDB-45883) (PDB 9CRU) | #2 Class 2 (EMDB-48826) (PDB 9N22) | #3 Class 3 (EMDB-45885) (PDB 9CRX) | #4 Class 4 (EMDB-49380) (PDB PNG2) | #5 Class 5 (EMDB-49522) (PDB 9NLU) | #6 Class 6 (EMDB-49524) (PDB 9NLW) | #7 Class 7 (EMDB-49526) (PDB 9NLY) | #8 Class 8 (EMDB-49527) (PDB 9NLZ) | #9 All Merge D1/D2 (EMDB-49528) (PDB 9NM1) | #10 Class 1 (EMDB-70472, no model deposited) | #11 Class 2 (EMDB-49801) (PDB 9NUD) | #12 Class 3 (EMDB-49802) (PDB 9NUE) | #13 Class 4 (EMDB-49824) (PDB 9NUZ) | #14 Class 1 (EMDB-49826) (PDB 9NV1) | #15 Class 2 (EMDB-49825) (PDB 9NV0) | #16 Class 1 (EMDB-49831) (PDB 9NV9) | #17 Class 2 (EMDB-49833) (PDB 9NVD) |
| **Data collection and processing** | | | | | | | | | | | | | | | | | |
| **Magnification** | 130,000× | 130,000× | 130,000× | 130,000× | 130,000× | 130,000× | 130,000× | 130,000× | 130,000x | 130,000x | 130,000x | 130,000x | 130,000x | 130,000x | 130,000x | 130,000x | 130,000x |
| **Camera** | Gatan K3 | Gatan K3 | Gatan K3 | Gatan K3 | Gatan K3 | Gatan K3 | Gatan K3 | Gatan K3 | Gatan K3 | Falcon 4i | Falcon 4i | Falcon 4i | Falcon 4i | Gatan K3 | Gatan K3 | Gatan K3 | Gatan K3 |
| **Voltage (kV)** | 300 | 300 | 300 | 300 | 300 | 300 | 300 | 300 | 300 | 300 | 300 | 300 | 300 | 300 | 300 | 300 | 300 |
| **Electron exposure (e–/Å2)** | 26.8 | 26.8 | 26.8 | 26.8 | 26.8 | 26.8 | 26.8 | 26.8 | 26.8 | 51.48 | 51.48 | 51.48 | 51.48 | 25.5 | 25.5 | 39.04 | 39.04 |
| **Defocus range (μm)** | -1.0 to -2.0 | -1.0 to -2.0 | -1.0 to -2.0 | -1.0 to -2.0 | -1.0 to -2.0 | -1.0 to -2.0 | -1.0 to -2.0 | -1.0 to -2.0 | -1.0 to -2.0 | -1.0 to -2.0 | -1.0 to -2.0 | -1.0 to -2.0 | -1.0 to -2.0 | -1.0 to -2.0 | -1.0 to -2.0 | -1.0 to -2.0 | -1.0 to -2.0 |
| **Pixel size (Å, super-res)** | 0.548 | 0.548 | 0.548 | 0.548 | 0.548 | 0.548 | 0.548 | 0.548 | 0.548 | 0.49 | 0.49 | 0.49 | 0.49 | 0.548 | 0.548 | 0.548 | 0.548 |
| **Exposure time (seconds)** | 4 | 4 | 4 | 4 | 4 | 4 | 4 | 4 | 4 | 6.93 | 6.93 | 6.93 | 6.93 | 4 | 4 | 4 | 4 |
| **Number of frames per exposure** | 80 | 80 | 80 | 80 | 80 | 80 | 80 | 80 | 80 | 2133 (EER fractionation of 42) | 2133 (EER fractionation of 42) | 2133 (EER fractionation of 42) | 2133 (EER fractionation of 42) | 80 | 80 | 80 | 80 |
| **Number of movies** | 12,439 | 12,439 | 12,439 | 12,439 | 12,439 | 12,439 | 12,439 | 12,439 | 12,439 | 14455 | 14455 | 14455 | 14455 | 7821 | 7821 | 13375 | 13375 |
| **Symmetry imposed** | C1 | C1 | C1 | C1 | C1 | C1 | C1 | C1 | C1 | C1 | C1 | C1 | C1 | C1 | C1 | C1 | C1 |
| **Final particle images (no.)** | 41,159 | 35,723 | 79,060 | 26,603 | 33,084 | 64,857 | 69,608 | 31,497 | 381,591 | 6,356 | 86,542 | 91,962 | 96,291 | 280,935 | 5,755 | 72,069 | 11,844 |
| **Global map resolution (Å)** | 3.89 | 3.91 | 3.73 | 4.61 | 4.29 | 3.73 | 3.75 | 4.13 | 3.4 | 10.88 | 3.38 | 3.36 | 3.18 | 2.99 | 7.83 | 3.60 | 4.70 |
| **Global map resolution range (Å): See supplementary information for resolution colored maps** | 3-7 | 3-7 | 3-7 | 3-7 | 3-7 | 3-7 | 3-7 | 3-7 | 2-4 | 9-12 | 2-6 | 2-7 | 2-4 | 2-3 | 4-8 | 3-6 | 4-7 |
| **FSC threshold** | 0.143 | 0.143 | 0.143 | 0.143 | 0.143 | 0.143 | 0.143 | 0.143 | 0.143 | 0.143 | 0.143 | 0.143 | 0.143 | 0.143 | 0.143 | 0.143 | 0.143 |
| **Model information** | | | | | | | | | | | | | | | | | |
| **Initial model used (PDB code)** | SNARE: 3B5N. Sec17: 1QQE/ AF-P32602-F1-v4. Sec18: AF-P18759-F1-v4 | SNARE: 3B5N. Sec17: 1QQE/ AF-P32602-F1-v4. Sec18: AF-P18759-F1-v4 | SNARE: 3B5N. Sec17: 1QQE/ AF-P32602-F1-v4. Sec18: AF-P18759-F1-v4 | SNARE: 3B5N. Sec17: 1QQE/ AF-P32602-F1-v4. Sec18: AF-P18759-F1-v4 | SNARE: 3B5N. Sec17: 1QQE/ AF-P32602-F1-v4. Sec18: AF-P18759-F1-v4 | SNARE: 3B5N. Sec17: 1QQE/ AF-P32602-F1-v4. Sec18: AF-P18759-F1-v4 | SNARE: 3B5N. Sec17: 1QQE/ AF-P32602-F1-v4. Sec18: AF-P18759-F1-v4 | SNARE: 3B5N. Sec17: 1QQE/ AF-P32602-F1-v4. Sec18: AF-P18759-F1-v4 | Sec18: AF-P18759-F1-v4 | Y20S EDTA Class 1 | *De novo* | *De novo* | *De novo* | *De novo* | *De novo* | 3J94 | 3J94 |
| **Bond RMSD (Å)** | 0.008 | 0.006 | 0.007 | 0.005 | 0.006 | 0.005 | 0.006 | 0.005 | 0.006 | No model deposited | 0.005 | 0.004 | 0.004 | 0.004 | 0.005 | 0.005 | 0.006 |
| **Angle RMSD (°)** | 1.322 | 0.952 | 1.212 | 0.904 | 1.111 | 0.964 | 0.839 | 0.852 | 1.051 | No model deposited | 1.261 | 0.957 | 0.793 | 0.771 | 1.077 | 1.169 | 1.228 |
| **Molprobity score** | 1.64 | 1.48 | 1.59 | 1.45 | 1.17 | 1.47 | 1.16 | 1.24 | 1.56 | No model deposited | 1.25 | 1.22 | 1.74 | 1.42 | 1.13 | 1.49 | 1.08 |
| **Clashscore, all atoms** | 5.61 | 3.73 | 5.26 | 3.55 | 1.58 | 4.65 | 1.79 | 1.72 | 4.16 | No model deposited | 2.84 | 2.30 | 3.93 | 3.57 | 1.20 | 1.08 | 1.53 |
| **Ramachandran. Favored/Allowed/Disfavored (%)** | 95.17 / 4.55 / 0.28 | 95.48 / 4.46 / 0.06 | 95.52 / 4.21 / 0.27 | 95.59 / 4.13 / 0.28 | 96.05 / 3.77 / 0.18 | 96.42 / 3.44 / 0.14 | 96.51 / 3.40 / 0.08 | 96.12 / 3.63 / 0.24 | 94.81 / 5.03 / 0.16 | No model deposited | 96.95 / 2.92 / 0.14 | 96.68 / 3.32 / 0.00 | 95.19 / 4.52 / 0.29 | 96.03 / 3.97 / 0.00 | 95.79 / 4.11 / 0.10 | 95.81 / 4.19 / 0.00 | 97.34 / 2.14 / 0.52 |
| **Rotamer outliers (%)** | 0.51 | 0.30 | 0.12 | 0.12 | 0.89 | 0.60 | 0.57 | 1.17 | 0.66 | No model deposited | 0.08 | 0.00 | 1.96 | 0.93 | 0.79 | 3.25 | 1.20 |
| **Cβ outliers (%)** | 0.00 | 0.02 | 0.00 | 0.00 | 0.00 | 0.00 | 0.00 | 0.02 | 0.00 | No model deposited | 0.04 | 0.24 | 0.00 | 0.00 | 0.00 | 0.00 | 0.03 |
| **CaBLAM outliers (%)** | 2.38 | 2.11 | 2.29 | 2.41 | 2.05 | 1.93 | 2.01 | 2.00 | 3.07 | No model deposited | 0.93 | 1.63 | 1.22 | 0.99 | 0.93 | 1.35 | 1.27 |
| **Model resolution (0.5)** | 4.20 | 4.55 | 4.60 | 5.50 | 4.50 | 4.30 | 4.40 | 5.0 | 3.50 | No model deposited | 3.50 | 3.60 | 3.40 | 3.20 | 9.25 | 3.80 | 5.20 |

# Reporting Summary

## Statistics

For all statistical analyses, confirm that the following items are present in the figure legend, table legend, main text, or Methods section.

| n/a | Confirmed | |
|---|---|---|
| ☐ | ☒ | The exact sample size (*n*) for each experimental group/condition, given as a discrete number and unit of measurement |
| ☐ | ☒ | A statement on whether measurements were taken from distinct samples or whether the same sample was measured repeatedly |
| ☐ | ☒ | The statistical test(s) used AND whether they are one- or two-sided *Only common tests should be described solely by name; describe more complex techniques in the Methods section.* |
| ☒ | ☐ | A description of all covariates tested |
| ☐ | ☒ | A description of any assumptions or corrections, such as tests of normality and adjustment for multiple comparisons |
| ☐ | ☒ | A full description of the statistical parameters including central tendency (e.g. means) or other basic estimates (e.g. regression coefficient) AND variation (e.g. standard deviation) or associated estimates of uncertainty (e.g. confidence intervals) |
| ☐ | ☒ | For null hypothesis testing, the test statistic (e.g. *F*, *t*, *r*) with confidence intervals, effect sizes, degrees of freedom and *P* value noted *Give P values as exact values whenever suitable.* |
| ☒ | ☐ | For Bayesian analysis, information on the choice of priors and Markov chain Monte Carlo settings |
| ☒ | ☐ | For hierarchical and complex designs, identification of the appropriate level for tests and full reporting of outcomes |
| ☒ | ☐ | Estimates of effect sizes (e.g. Cohen's *d*, Pearson's *r*), indicating how they were calculated |

*Our web collection on statistics for biologists contains articles on many of the points above.*

## Software and code

Policy information about availability of computer code

| Data collection | SerialEM 4.1, Molecular Devices SoftMax Pro |
|---|---|
| Data analysis | Tmaven, Byonic v5.x.x, AlphaFold2, ChimeraX 1.8, Coot 0.9, CryoSPARC 4.X, CTFFIND4, ImageJ, ISOLDE, MolProbity, MotionCorr2, PHENIX 1.2.X, GraphPad Prism, RELION 3, custom Python 3.X scripts using packages numpy, scipy, scikit-learn, biopython Bio.PDB, pandas, and matplotlib. |

For manuscripts utilizing custom algorithms or software that are central to the research but not yet described in published literature, software must be made available to editors and reviewers. We strongly encourage code deposition in a community repository (e.g. GitHub). See the Nature Portfolio guidelines for submitting code & software for further information.

## Data

Policy information about availability of data

All manuscripts must include a data availability statement. This statement should provide the following information, where applicable:
- Accession codes, unique identifiers, or web links for publicly available datasets
- A description of any restrictions on data availability
- For clinical datasets or third party data, please ensure that the statement adheres to our policy

Accession codes available upon publication. No restrictions on data availability.

# Research involving human participants, their data, or biological material

Policy information about studies with <u>human participants or human data</u>. See also policy information about <u>sex, gender (identity/presentation), and sexual orientation</u> and <u>race, ethnicity and racism</u>.

| | |
|---|---|
| Reporting on sex and gender | Does not apply |
| Reporting on race, ethnicity, or other socially relevant groupings | Does not apply |
| Population characteristics | Does not apply |
| Recruitment | Does not apply |
| Ethics oversight | Does not apply |

Note that full information on the approval of the study protocol must also be provided in the manuscript.

# Field-specific reporting

Please select the one below that is the best fit for your research. If you are not sure, read the appropriate sections before making your selection.

☒ Life sciences    ☐ Behavioural & social sciences    ☐ Ecological, evolutionary & environmental sciences

For a reference copy of the document with all sections, see <u>nature.com/documents/nr-reporting-summary-flat.pdf</u>

# Life sciences study design

All studies must disclose on these points even when the disclosure is negative.

| | |
|---|---|
| Sample size | Disassembly assays described in this study were performed with at least 3 samples for each group, which is consistent with sample sizes commonly used in similar studies in the field. Single molecule FRET assays have sample size (number of traces analyzed) for each condition in main figure panels. |
| Data exclusions | Cryo-EM reconstructions with resolutions worse than 9 Å were excluded due to challenges associated with accurate model building and subsequent comparison with other reconstructions. For single molecule FRET experiments, traces were automatically excluded prior to analysis by automated tMaven filtering that had no manual intervention or parameter tuning. |
| Replication | In-vitro assays were performed in different batches with different mixes of in-vitro proteins to ensure independence of each assay. For in-vivo mass-spec, uninduced and induced samples were always processed simultaneously and only differed by galactose induction in cell media prior to processing. |
| Randomization | N/A |
| Blinding | N/A |

# Reporting for specific materials, systems and methods

We require information from authors about some types of materials, experimental systems and methods used in many studies. Here, indicate whether each material, system or method listed is relevant to your study. If you are not sure if a list item applies to your research, read the appropriate section before selecting a response.

## Materials & experimental systems

| n/a | Involved in the study |
|---|---|
| ☐ | ☒ Antibodies |
| ☐ | ☒ Eukaryotic cell lines |
| ☒ | ☐ Palaeontology and archaeology |
| ☒ | ☐ Animals and other organisms |
| ☒ | ☐ Clinical data |
| ☒ | ☐ Dual use research of concern |
| ☒ | ☐ Plants |

## Methods

| n/a | Involved in the study |
|---|---|
| ☒ | ☐ ChIP-seq |
| ☒ | ☐ Flow cytometry |
| ☒ | ☐ MRI-based neuroimaging |

# Antibodies

| | |
|---|---|
| Antibodies used | Pierce HA-Tag IP/Co-IP kit |
| Validation | IP of HA-tagged SRF and HA-tagged Pak1 were performed. For HA-Sec18, IP was confirmed independently by western blot with a different HA antibody to confirm enrichment of HA-Sec18. |

# Eukaryotic cell lines

Policy information about cell lines and Sex and Gender in Research

| | |
|---|---|
| Cell line source(s) | S288C Saccharomyces cerevisiae |
| Authentication | 18S ribosomal RNA sequencing and 28S ribosomal RNA sequencing |
| Mycoplasma contamination | N/A |
| Commonly misidentified lines (See ICLAC register) | N/A |

# Plants

| | |
|---|---|
| Seed stocks | N/A |
| Novel plant genotypes | N/A |
| Authentication | N/A |

