## [Peer Review File · Nature Structural & Molecular Biology]

SNARE disassembly requires Sec18/NSF side-loading

Corresponding Author: Dr Axel Brunger

Version 0:

Decision Letter:

11th Oct 2024

Dear Dr. Brunger,

Thank you again for submitting your manuscript "Sec18 side-loading is essential for universal SNARE recycling across cellular contexts". Please accept my apologies for the very long delay in reaching a decision on this manuscript. I am afraid we have had a reduced editorial team for the past month, due to illness and travel. I'm writing to let you know that we have decided to send your manuscript for peer review.

I am re-opening the manuscript submission link for you to resubmit your manuscript with all the associated files needed for the peer review process directly to our system, at your convenience. Please also include the study 'NSF converts syntaxin nanodomains to a priming-ready state' as a supplementary file, for reference to the potential referees. Please see below for details regarding the required materials. Please follow the link at the bottom of this email to upload the documents.

We want to ensure that the methods and statistics reporting in our papers are of the highest quality. To that end, we ask authors to fill out a Reporting Summary that collects information on experimental design and reagents, as well as an editorial Policy Checklist, which confirms compliance with our editorial policies, including the declaration of Competing Interests. If your paper includes ChIP-seq, flow cytometry or MRI data, we ask you take special care to complete those sections of the Reporting Summary as this data will aid greatly in the review of your manuscript.

These documents can be found by following the links below:

Reporting Summary:

Editorial Policy Checklist: <https://www.nature.com/documents/nr-editorial-policy-checklist.pdf>

Please be aware of our guidelines on digital image standards.

Please note we require official wwPDB validation reports for newly described atomic structures, as noted in the policy checklist. We also request that authors provide cryo-EM maps, half-maps and models, to help the reviewers in assessing the work. We recommend the use of figshare integration into our systems, which allows for provision of anonymous access links for the referees (<https://www.springernature.com/gp/authors/research-data/figshare-integration>). Alternatively, please upload .zip folders directly with the submission. To ensure the ease of reviewer access to the data, please specify in the Data Availability section, where the files can be found (provide a figshare link or direct the reader to the manuscript files).

Please use the link below to submit the files. **Please also remember to move forward all other files associated with this version of the paper.**

Link Redacted

Sincerely,

Carolina Perdigoto, PhD
Chief Editor
Nature Structural & Molecular Biology
orcid.org/0000-0002-5783-7106

Version 1:

Decision Letter:

6th Dec 2024

Dear Dr. Brunger,

Thank you again for submitting your manuscript "Sec18 side-loading is essential for universal SNARE recycling across cellular contexts". I apologize for the delay in responding, which resulted from the difficulty in obtaining suitable referee reports. Nevertheless, we now have comments (below) from the 4 reviewers who evaluated your paper. In light of those reports, we remain interested in your study and would like to see your response to the comments of the referees, in the form of a revised manuscript.

You will see that while reviewers appreciate the results, they raise several concerns which will need to be addressed in a revision.

Specifically, please respond to all technical comments regarding the cross-linking data from referees #1, #3 and #4, regarding the validity and confidence of cross-links, as well as how this data fits with the model and the cryo-EM data. Additionally, we recommend deposition of data in public repositories prior to publication. We also agree with reviewer #1 and #4 that further elaboration on the side-loading mechanism, and the details of ring opening would strengthen the manuscript. Furthermore, you will see that reviewer #2 requests clarification of hexamer/heptamer conversion during hydrolysis, and we ask that the point is addressed.

Please be sure to address/respond to all concerns of the referees in full in a point-by-point response and highlight all changes in the revised manuscript text file. If you have comments that are intended for editors only, please include those in a separate cover letter.

We expect to see your revised manuscript within 3-4 months. If you cannot send it within this time, please contact us to discuss an extension; we would still consider your revision, provided that no similar work has been accepted for publication at NSMB or published elsewhere.

Please reach out if you'd like to discuss a revision plan.

Reporting Summary:
<https://www.nature.com/documents/nr-reporting-summary.pdf>

When submitting the revised version of your manuscript, please pay close attention to our <https://www.nature.com/nature-portfolio/editorial-policies/image-integrity> Digital Image Integrity Guidelines and to the following points below:

-- that unprocessed scans are clearly labelled and match the gels and western blots presented in figures.

-- that control panels for gels and western blots are appropriately described as loading on sample processing controls
-- all images in the paper are checked for duplication of panels and for splicing of gel lanes.

Please note that all key data shown in the main figures as cropped gels or blots should be presented in uncropped form, with molecular weight markers. These data can be aggregated into a single supplementary figure item. While these data can be displayed in a relatively informal style, they must refer back to the relevant figures. These data should be submitted with the final revision, as source data, prior to acceptance, but you may want to start putting it together at this point.

Data availability: this journal strongly supports public availability of data. All data used in accepted papers should be available via a public data repository, or alternatively, as Supplementary Information. If data can only be shared on request, please explain why in your Data Availability Statement, and also in the correspondence with your editor. Please note that for some data types, deposition in a public repository is mandatory - more information on our data deposition policies and available repositories can be found below:

<https://www.nature.com/nature-research/editorial-policies/reporting-standards#availability-of-data>

Link Redacted

Sincerely,

Katarzyna Ciazynska, PhD
(she/her)
Senior Editor
Nature Structural & Molecular Biology
<https://orcid.org/0000-0002-9899-2428>

Reviewers' Comments:

Reviewer #1 (Remarks to the Author):

This manuscript by Khan et al. investigates the mechanism of SNARE complex disassembly by Sec18, the yeast homolog of NSF. The authors employed a cross-linking strategy to map interactions between different SNARE proteins and various domains of Sec18. They subsequently determined cryo-electron microscopy (cryo-EM) structures of the SNARE complex-loaded yeast 20S complex (Figures 2 and 3) and substrate-free Sec18 under different nucleotide hydrolysis conditions (Figures 5 and 6). Based on these results, the authors propose that both the D1 and D2 rings of Sec18 open to load or unload the SNARE complex substrate via a "side-loading" mechanism.

Notably, this study presents the first high-resolution structure of the yeast 20S complex. While the yeast 20S complex shares an overall structure and disassembly mechanism with the neuronal 20S complex from mammalian species, the proposed side-loading mechanism is particularly intriguing. It offers potential answers to unresolved questions, such as how the N-terminal domain is accommodated within the 20S complex. This is significant given prior observations that SNARE complex disassembly occurs through a single major energy barrier rather than a processive mechanism.

While the data and conclusions are interesting, several issues need to be addressed before I can recommend this manuscript for publication in Nature Structural & Molecular Biology.

Major Points

Crosslinking Data Validation: The cross-linking data is intriguing but seems to require significant manual interpretation to distinguish high-confidence cross-links from middle- or low-confidence ones. Furthermore, it is unclear whether the cross-linking data align with the cryo-EM structure in Figure 2 and 3. The authors should clarify whether the observed cross-links between Sso1 and Sec18 (as well as Snc2 and Sec18) are consistent with the contact sites resolved in the cryo-EM structure.

Mechanistic Details of the Side-Loading Mechanism: The side-loading mechanism is compelling but demands a synchronized opening of both the D1 and D2 rings. Since the D2 ring is primarily responsible for maintaining the hexameric structure and exhibits minimal ATPase activity, it remains unclear how it could open while still bound to ATP. Additionally, the manuscript lacks sufficient discussion on how the D1 and D2 rings communicate to achieve coordinated opening. These aspects require further elaboration.

Minor Points

1. The domain structure of Sso1 in Figure 1A should be revised. The Habc domain should be positioned N-terminal to the SNARE motif.
2. In Figure 6G, protein labels should be specified to enhance readability.
3. Case usage for terms such as SEC18 and Habc is inconsistent across the figures and should be standardized.
4. In line 286, the cited reference to Figure 5C appears incorrect; it should likely be Figure 5E.
5. Figure organization could be improved. For instance, the probability graph in Figure 4D could benefit from unified Y-axis scaling to avoid separation.

Reviewer #2 (Remarks to the Author):

Using crosslinking mass-spectrometry and cryo-EM, Kahn et al. show that N-terminal domains of SNAREs can interact with the D2 ring of NSF/Sec18. Detailed cryo-EM studies/data using ATP-hydrolyzing and non-hydrolyzing conditions suggest a model in which both the D1 and D2 rings of NSF/Sec18 laterally open to initially load assembled cis-SNARE complexes and to subsequently release disassembled SNARE complexes. The data are overall convincing and the proposed molecular mechanism of NSF/Sec18 action profoundly forwards our understanding of membrane fusion and will be of great interest for a broad readership. A surprising finding is the presence of heptameric Sec18-1 under hydrolyzing conditions. The γ 20S complex (non-hydrolyzing conditions) seems to contain only hexameric Sec18 (Figure 2). Please clarify. How would hydrolyzing conditions convert the Sec18 hexamers into heptamers? Heptamers can also be generated in higher amounts in an in vitro Sec18 disassembly/reassembly process. The physiological conversion of hexamers into heptamers remains puzzling, but this issue could be resolved in future follow up studies. Nevertheless, the authors should consider to directly compare the activities of hexameric and heptameric Sec18 in their in vitro fusion assay, if possible. The authors may also comment on the role of these Sec18 hexamer/heptamer states in their discussion. In addition, the following minor technical points should be addressed:

- Figure 2: It would be helpful to indicate which amino acid stretch of Sso1 interacts with the Y315 tyrosine region of the D1 protomers.
- Figure 3D: Nucleotides are mentioned in the legend, but they are not detectable in the figure
- Figure 4B and C, FRET assay: Are the assignments of donor = violet and acceptor = orange correct? Briefly state in the Methods section which fluorophore pairs were used (Oregon green and Alexa Fluor 568 for neuronal SNARE complex?).
- In general please check the Methods section for completeness; e.g. crosslinking assay: please mention pH of Tris buffer and which beads were used.
- Extended data Figure 5B: Please assign molecular weights to the marker lane.
- Extended data Fig. 5C legend: Briefly mention which reactants were increased in this assay (SNARE complexes?)
- Extended data Figure 6E, figure legend: Does the figure show slices of Sec18 in the γ 20S complex or of isolated Sec18?
- In addition to the mass spectrometry (extended data, Figure 6G) please mention in the text the (double) crosslinking efficiency for the SSO1 Habc domain. This is relevant for the results shown in Figure 4 G,H.

- Line 285: The structure shown in Figure 6B likely refers to the structure in Figure 5E (instead of Figure 5C). Please clarify.
- Line 293: Please check if referring to Figure 4A is correct (more likely Figure 5C). (In general, it would be helpful to include on pages containing individual figures the actual figure number.)

Reviewer #3 (Remarks to the Author):

In this interesting study, a combination of experimental approaches, centered around sm cryo-EM, is used to unravel the mechanism by which the AAA-protein NSF (or more precisely its yeast orthologue Sec18p) dissociates SNARE complexes. NSF/Sec18p are hexamers consisting of three domains, the ATP-binding D2 and D1 domains forming hexameric rings, and the mobile N-domains. The D2 domain subunits are thought to remain in the ATP-bound state during the catalytic cycle, being responsible for oligomerization, whereas the D1 domains hydrolyze ATP. In previous work, the Brunger lab solved the structure of the arrested NSF/SNARE/alphaSNAP complex, showing that the assembled SNARE complex extends into the central pore of the D1 domain, with part of SNAP-25 extending to the outside, suggesting sideways opening of the D1 ring during substrate release.

In the present work, the analysis of the SNARE disassembly mechanism is taken further, resulting in a first, albeit still somewhat tentative model for the catalytic cycle of the NSF/Sec18p system. First, the authors use in-vivo X-linking with a membrane-permeable homo-bifunctional X-linker in yeast, followed by isolation of Sec18p complexes, resulting in a high yield of Sec18p-SNARE X-links that connect not only the D1 but also the D2 domains to SNAREs. The authors then move on and solve the structure of the yeast exocytotic 20S complex. They observe that in the complex the N-terminal 3-helix bundle (Habc-domain) of the syntaxin orthologue Sso1 is positioned on the distal surface of the D2-domain, in a fully folded form. Moreover, apparently the resolution also suffices to identify the linker connecting the Habc-domain to the SNARE motif threading through the pore in the center of the hexameric D2-ring. The authors then use two independent approaches suggesting that the Habc domain does not unfold during the catalytic cycle, leading the authors to conclude that the only way the linker can become threaded through the central pore is by a “clamp-loader” mechanism. To clarify how the SNAREs are released after disassembly, the authors trigger disassembly by adding Mg-ions to start the ATPase and then vitrify the sample after a few s to look at intermediates. Interestingly, most intermediates are free of substrate, with some of the class averages showing either the D1 or both D1 and D2-rings with a split in the ring, resulting in a model according to which both rings open for substrate loading and then open again for substrate release after ATP-hydrolysis, associated with conformational changes (flattening) in the D1 ring arrangements.

The manuscript is well illustrated and contains a lot of data, and although some of the conclusions will need further corroboration in the future, the data as presented support the proposed mechanism for the catalytic cycle of SNARE disassembly. In particular, the finding that at least the syntaxin homologue is threaded through both rings, with the folded Habc-domain bound to the top of the membrane-distal D2 ring, is novel and exciting. Nevertheless, I have several questions that I would like the authors to address during revision (not necessarily requiring additional experiments):

1. In the x-linking experiments, the pattern has an almost random nature, with all parts of the SNAREs being cross-linked to all domains of Sec18p. In addition to the x-links between the N-terminal domains and the D2 subunits that are supporting the model, there are several X-links between the SNARE-motifs and the D2 domain. Obviously, there is always some noise in such data, and considering further that in such experiments X-links can also occur during diffusional collisions, such data, particularly with respect to intermolecular X-links need to be interpreted with caution, but they weaken the confidence in the specificity of these data. Did the authors analyze all high- and medium confidence X-links between the SNARE and the Sec18 domains, and to which extent do they confirm vs. contradict the model?
2. In the SM fluorescence experiments I am a bit puzzled that there is absolutely no change in fluorescence during the catalytic cycle when the Habc domains are labeled. Did the authors monitor lifetime changes? It is hard to believe that during the predicted “clamp-loading” and the transient bindings of the domain onto the surface of Sec18 (which is expected to change the local environment of the fluorophore quite substantially) quantum yield/lifetime are not affected at all.
3. Habc crosslinking experiments shown in Fig. 4 G and H: Did the authors verify that the X-linking had indeed worked, e.g. by showing that the domain is resistant to unfolding?
4. It is unfortunate that in the experiments attempting to trap intermediate states during substrate release the substrates are not visible in the splits of the D1 or D2 rings. Are the class 2 and class 3 structures showing the splits indeed only detectable after triggering of hydrolysis? Are they dependent on the presence of substrate, i.e. is it possible to exclude that such splits also occur, perhaps randomly, in the absence of substrates and Sec17?
5. It is known that NSF also disassembles truncated SNARE complexes lacking Habc or other structured N-terminal domains. Do the authors assume that in such cases the D2-ring does not participate? Conversely, what would happen if the linker between the SNARE domain and the Habc domain is shortened so that it cannot span the D2-ring anymore – would such complexes be resistant to disassembly?

Reviewer #4 (Remarks to the Author):

In the present work, Khan et al. use cross-linking mass spectrometry, cryo-EM imaging, and bulk and single molecule approaches to report on the mechanism by which Sec18, an NSF homolog, engages and unfolds client SNARE complexes. The findings are used to argue that SNARE unfolding is not necessarily initiated by first threading a target chain into the translocation pore of the D1 AAA+ ring and then passing this substrate to the pore of the adjoining D2 ring of Sec18; instead, it is suggested that the two rings may cooperatively split apart, allowing a chain to enter the translocation pore laterally from the side before pulling and complex disruption takes place.

The data appear of high technical quality and the conclusions plausible. The work is accessible to a broad audience and is likely to be an important advance for the membrane fusion, vesicle trafficking, and AAA+ ATPase fields, as it provides a compelling explanation for how Sec18/NSF can unfold SNAREs in which access to a free SNARE chain is blocked by folded domains on both ends while also expanding the number of AAA+ systems that use side loading mechanisms for substrate engagement. Pending the resolution of a few questions and issues, publication in NSMB would seem warranted.

Primary comments:

Lines 154-155. Please show density for all stated nucleotide assignments.

Line 172, Section on substrate-free Sec18 structures. Why were Sec18 oligomers dissociated into monomers and then reassembled as opposed to just using native, substrate-free oligomers? How can one know that this workflow isn't giving rise to improperly assembled oligomers? Is it just because of the activity seen in ED Fig. 6H-J? Is this level of activity the same as seen for native Sec18 that hasn't been disassembled?

Fig. 3D. Please show a zoomed-in view of the Sso1 density in the D2 ring.

Fig. 4G and H. Why is the signal for the doubly crosslinked Habc so much weaker than for the uncrosslinked? This signal difference makes the relative fraction of disassembly comparison questionable.

Fig. 6. What promotes ring opening? Could a closed ring engage the SNARE substrate through the N-domains, which might then trigger opening? Can open rings be seen if Sec18 is mixed with a SNAP/SNARE complex lacking the Sso1 Habc domain and either some or all of the linker?

Fig. 6. What is the role of the heptamer?

ED Fig. 6D. How was it confirmed from the native gel that the major band is the heptamer and the minor species a hexamer? Could an alternative method like mass photometry be used to show this?

Minor points:

Lines 56-57 and 64-65 are repetitive.

Line 72. Change to "were" bound.

Lines 105-107. There is no Supplementary Figure 1 as referenced, nor does ED Fig. 1 reflect the information referenced in this statement. Are the SAFE/GO data available as a downloadable supplemental table?

ED Figs. 2C and 5a - should the colors of Snc1 and Sso1 be swapped to match those of the main text figures?

Line 181. Should this read "that precedes it"? According to Fig. 2A, the Sso1 Habc domain is C-terminal to the linker?

Lines 364-369 - side loading also occurs for the AAA+ hexameric helicase MCM2-7, the RecA-family DnaB and Rho helicases, and the AAA+ ORC and DnaC replicative helicase loaders.

Fig. 2. It is difficult to see the blue and red nucleotides against the pink and lightblue protein chains. Can these be more clearly distinguished?

Version 2:

Decision Letter:

Our ref: NSMB-A49750B

3rd Feb 2025

Dear Dr. Brunger,

Thank you for submitting your revised manuscript "Sec18 side-loading is essential for universal SNARE recycling across cellular contexts" (NSMB-A49750B). It has now been seen by the original referees and their comments are below. The reviewers find that the paper has improved in revision, and therefore we'll be happy in principle to publish it in Nature Structural & Molecular Biology, pending minor revisions to satisfy the referees' final requests and to comply with our editorial and formatting guidelines.

We are now performing detailed checks on your paper and will send you a checklist detailing our editorial and formatting requirements in about 2-3 weeks. Please do not upload the final materials and make any revisions until you receive this additional information from us.

Sincerely,

Katarzyna Ciazynska, PhD
(she/her)
Senior Editor
Nature Structural & Molecular Biology
<https://orcid.org/0000-0002-9899-2428>

Reviewer #1 (Remarks to the Author):

The authors have addressed the concerns raised during the previous review effectively. In particular, the addition of the subsection discussing the opening of the D1 and D2 rings, along with Extended Data Figure 8, is a very welcome addition, convincingly suggesting that the opening of D1 and D2 rings would be a largely thermally driven process. I have only a few minor comments that could enhance the manuscript further before its final publication.

1. In Figure 3c, the shaded structures are nearly invisible, which may hinder reader comprehension. I suggest adjusting the transparency to make the structures more discernible. Alternatively, if space is a concern, these structures could be removed entirely.

2. Regarding the biochemical characteristics of the Sec18 heptamer presented in Figure 7, it remains unclear whether this heptameric conformation facilitates the formation of the 20S complex. The cryo-EM structures of selectively purified 20S complexes shown in Figures 2 and 6 depict only hexameric conformations. Although vacuole fusion assays were performed, with fusion reaching approximately 10% of the saturated signal (as shown in Extended Data Fig. 6J), it is conceivable that the hexameric conformations were predominantly active, while the heptameric conformers remained largely inactive. In the Discussion section (lines 404-421), where the implications of the heptameric states are considered, it would be beneficial for the authors to clarify whether these heptameric conformers are indeed competent for 20S complex formation.

3. On line 391, please clarify that the N-terminal domains referenced pertain to SNARE proteins, not those of Sec18.

Reviewer #2 (Remarks to the Author):

The combined XL-MS and cryo-EM approach provides important and novel structural and mechanistic insights, of how distinct members of the AAA-ATPase family load and unload their substrates. The authors have appropriately addressed my previous concerns/comments.

Reviewer #3 (Remarks to the Author):

During revision, the authors have provided detailed responses to each of the points I have raised in my original review, which are comprehensive and clear. For these reasons, I support publication of the manuscript in its present form.

Reviewer #4 (Remarks to the Author):

The authors have satisfactorily addressed all comments raised in the review. There are no additional concerns regarding this interesting set of findings.

Responses to reviewers' comments

The reviewer's comments are italicized, and our responses are in blue. The manuscript with tracked changes is included as a "additional review material".

Responses to Reviewer 1

"[T]he proposed side-loading mechanism is particularly intriguing. It offers potential answers to unresolved questions, such as how the N-terminal domain is accommodated within the 20S complex. This is significant given prior observations that SNARE complex disassembly occurs through a single major energy barrier rather than a processive mechanism."

We thank reviewer 1 for their kind words.

"Crosslinking Data Validation: The cross-linking data is intriguing but seems to require significant manual interpretation to distinguish high-confidence cross-links from middle- or low-confidence ones."

We apologize for not clarifying the crosslinking data analysis sufficiently. We have expanded the methods (lines 689-705) to define the quantitative cutoffs (false discovery rate, precursor mass error, and peptide length) used to assess crosslink quality quantitatively.

"It is unclear whether the cross-linking data align with the cryo-EM structure in Figure 2 and 3. The authors should clarify whether the observed cross-links between Sso1 and Sec18 (as well as Snc2 and Sec18) are consistent with the contact sites resolved in the cryo-EM structure."

We thank the reviewer for the excellent suggestion. We have added the high-quality intra- and interprotomer Sec18 crosslinks in **Figure 3**, which shows that these crosslinks agree with our cryo-EM structure. The critical D2 to Sec18-Sso1 crosslink is now shown in **Figure 4A**, illustrating how it matches the cryo-EM structure. The Sec18-snc2 crosslinks are consistent with current and previous data, which we have noted in the text (lines 167-173).

"Mechanistic Details of the Side-Loading Mechanism: The side-loading mechanism is compelling but demands a synchronized opening of both the D1 and D2 rings. Since the D2 ring is primarily responsible for maintaining the hexameric structure and exhibits minimal ATPase activity, it remains unclear how it could open while still bound to ATP. Additionally, the manuscript lacks sufficient discussion on how the D1 and D2 rings communicate to achieve coordinated opening. These aspects require further elaboration."

We thank the reviewer for the excellent question. We have conducted a new conformational analysis comparing the closed and open ring cryo-EM classes of Sec18 in the hydrolyzing condition cryo-EM data, and the results are presented in the new **Extended Data Figure 8**, a new section on “Mechanistic details of ring opening” (lines 343-365), and in the Discussion (lines 404-432). In short, simple backbone motions of a limited number of residues in flexible regions of the protomers lead to rigid body motions of sub-domains, allowing for this large opening to form. As the reviewer notes, the D2 protomers are largely structural (*i.e.*, there is no observable hydrolysis in the D2 protomers), so this process is likely driven by hydrolysis and nucleotide exchange by the D1 protomers. Presumably, the split interfaces of D1 and D2 are weaker than the in-ring interfaces due to slight asymmetry or misalignment, and cycling by D1 provides sufficient energy to disrupt the ATP-engaged D2 interprotomer interface. In all states where Sec18 is substrate-free, the D1 protomers all contain ADP, not ATP. Regarding the overall mechanism, we believe that the open and closed ring states are sampled regularly by Sec18 without hydrolysis or the presence of substrate.

“The domain structure of Sso1 in Figure 1A should be revised. The Habc domain should be positioned N-terminal to the SNARE motif.”

We have all the SNAREs positioned C to N terminally compared to Sec18, which is N→C for ease of reading. However, we found a small error in **Figure 2A**, which we have corrected.

“In Figure 6G, protein labels should be specified to enhance readability.”

We thank the reviewer for the excellent suggestion and added labels.

Case usage for terms such as SEC18 and Habc is inconsistent across the figures and should be standardized.

Thank you for catching these inconsistencies. We have fixed these issues.

In line 286, the cited reference to Figure 5C appears incorrect; it should likely be Figure 5E.

Thank you for catching this error. We have fixed it.

Responses to Reviewer 2

“The data are overall convincing and the proposed molecular mechanism of NSF/Sec18 action profoundly forwards our understanding of membrane fusion and will be of great interest for a broad readership.”

We thank reviewer 2 for their kind words.

“A surprising finding is the presence of heptameric Sec18-1 under hydrolyzing conditions. The y20S complex (non-hydrolyzing conditions) seems to contain only hexameric Sec18 (Figure 2). Please clarify.”

We agree with reviewer 2 that the presence of heptameric Sec18 under hydrolyzing conditions is surprising; we were also initially surprised by this result. Please see lines 404-419 in the discussion where we now thoroughly discuss this point.

“How would hydrolyzing conditions convert the Sec18 hexamers into heptamers?”

We observed heptameric states of Sec18 (**Figure 6C**) and NSF (**Extended Data Figure 7A**), split-open hexameric states of Sec18 (**Figure 6E**) and NSF (<https://www.biorxiv.org/content/10.1101/2024.10.11.617886v1>), and a transition between split-hexamer and heptamer of Sec18 (**Figure 6D**). Hydrolysis likely accelerates this transition for two related reasons. First, while ring splitting itself is hydrolysis-independent, hydrolysis is associated with a large-scale conformational change in the ATPase rings and likely increases the frequency of sampling the open state. Second, we have not observed the heptamer in a substrate-engaged state. Moreover, it seems unlikely that a heptamer could be accommodated given the changes heptamerization induces in the N- and D1- layers of the oligomer (an increased diameter might interfere with α -SNAP recognition of substrate and substrate engagement in the pore). A heptamer is enriched under hydrolyzing conditions either in the absence of SNARE substrate or in which a significant fraction of SNARE complexes have been disassembled—the hexamer samples the open state and the seventh protomer enters in the absence of SNARE substrate. We have added a new Extended Data Figure 8, which describes a conformational analysis explaining how rigid body motions in key residues sample this split-hexamer, a new section entitled “Mechanistic details of ring opening”, and expanded the discussion (lines 404-421).

“Nevertheless, the authors should consider to directly compare the activities of hexameric and heptameric Sec18 in their in vitro fusion assay, if possible. “

Thank you for the excellent suggestion. We did attempt to do so. However, we could not purify Sec18 under non-hydrolyzing conditions (unlike NSF). Because Sec18 is only purifiable in hydrolyzing conditions, all pure Sec18 in the absence of substrate is also a mixture of heptamers and hexamers that we could not separate with several biophysical techniques (SEC, sedimentation, etc).

“The authors may also comment on the role of these Sec18 hexamer/heptamer states in their discussion.”

This is an excellent point. We have added a new section in the Discussion addressing the interesting points raised (lines 404-419).

“It would be helpful to indicate which amino acid stretch of Sso1 interacts with the Y315 tyrosine region of the D1 protomers.”

Done.

“Figure 3D: Nucleotides are mentioned in the legend, but they are not detectable in the figure”

We thank the reviewer for catching this issue. We have fixed the legend.

“Figure 4B and C, FRET assay: Are the assignments of donor = violet and acceptor = orange correct? Briefly state in the Methods section which fluorophore pairs were used (Oregon green and Alexa Fluor 568 for neuronal SNARE complex?).”

We apologize for any confusion; both fluorophores are Cy3 and Cy5 in both experiments. We have revised the colors in the figure to clarify this.

“In general please check the Methods section for completeness; e.g. crosslinking assay: please mention pH of Tris buffer and which beads were used.”

In the Methods, we now describe the crosslinking buffer (pH 7) and the spin column used to remove excess dye (Zeba Spin Column).

“Extended data Figure 5B: Please assign molecular weights to the marker lane.”

Thank you for catching this issue. We added molecular weights to the marker lane.

Extended data Fig. 5C legend: Briefly mention which reactants were increased in this assay (SNARE complexes?)

We have clarified this in the legend now. It is indeed “SNARE complexes”.

Extended data Figure 6E, figure legend: Does the figure show slices of Sec18 in the Y20S complex or of isolated Sec18?

We have clarified this in the legend now. The figure shows the isolated Sec18 molecule.

“ In addition to the the mass spectrometry (extended data, Figure 6G) please mention in the text the (double) crosslinking efficiency for the SSO1 Habc domain. This is relevant for the results shown in Figure 4 G,H.”

We did not find any mass peaks corresponding to uncrosslinked Sso1 in the sample, and we have now stated this explicitly.

“Line 285: The structure shown in Figure 6B likely refers to the structure in Figure 5E (instead of Figure 5C). Please clarify.”

Thank you. We have fixed the issue.

“Line 293: Please check if referring to Figure 4A is correct (more likely Figure 5C). (In general, it would be helpful to include on pages containing individual figures the actual figure number.)”

Thank you. We have fixed the issue.

Responses to Reviewer 3

“The manuscript is well illustrated and contains a lot of data, and although some of the conclusions will need further corroboration in the future, the data as presented support the proposed mechanism for the catalytic cycle of SNARE disassembly. In particular, the finding that at least the syntaxin homologue is threaded through both rings, with the folded Habc-domain bound to the top of the membrane-distal D2 ring, is novel and exciting. Nevertheless, I have several questions that I would like the authors to address during revision (not necessarily requiring additional experiments)”

We thank reviewer 3 for their kind words.

Did the authors analyze all high- and medium-confidence X-links between the SNARE and the Sec18 domains, and to which extent do they confirm vs. contradict the model?

This is an excellent point by Reviewer 3; we have now indicated the crosslinks in the revised **Figures 3 and 4**, illustrating the agreement with our cryo-EM structures. In particular, we depict the high-quality crosslink between D2 and Habc in Figure 4. The other high-quality crosslinks are consistent with the cryo-EM structures, including the cross-links between Snc2 and Sec18 that support our current loading and release model, which we have noted in the text now on lines 167-173 and 280-282.

Did the authors monitor lifetime changes? It is hard to believe that during the predicted “clamp-loading” and the transient bindings of the domain onto the surface of Sec18 (which is expected to change the local environment of the fluorophore quite substantially) quantum yield/lifetime are not affected at all.

The label locations are at positions 35 and 105 (revised **Figure 5A**), which, based on our cryo-EM structures, are at the tips of the α -helices that are positioned away from the D2 pore and the Sec18 complex. Thus, one would expect that 20S complex formation does not affect the quantum yield and conformational dynamics of the dyes, which is indeed what we observe. We have added this point to the Methods (lines 843-849).

Habc crosslinking experiments shown in Fig. 4 G and H: Did the authors verify that the X-linking had indeed worked, e.g. by showing that the domain is resistant to unfolding?

We thank the reviewer for the excellent question. We tested our unnatural amino acid crosslinking efficiency by performing LC-ESI/MS to confirm that the double crosslinking had occurred. We were able to confirm this since the product of a double crosslinking reaction would lead to a 28762 Dalton product, which we were able to observe (28762.1 experimentally) in **Extended Data Figure 5G**. We did not detect a mass peak for the uncrosslinked product.

Are the class 2 and class 3 structures showing the splits indeed only detectable after triggering of hydrolysis? Are they dependent on the presence of substrate, i.e. is it possible to exclude that such splits also occur, perhaps randomly, in the absence of substrates and Sec17?

These splits do occur in the absence of substrate. In the revised **Figure 7B**, we show the split Sec18 structure that we could resolve in a cryo-EM dataset for Sec18 without Sec17 or SNARE substrate. Given that the splits occur after triggering hydrolysis with substrate and in the absence of substrate, these observations support our models of side-loading and side-release.

It is known that NSF also disassembles truncated SNARE complexes lacking Habc or other structured N-terminal domains. Do the authors assume that in such cases the D2-ring does not participate? Conversely, what would happen if the linker between the SNARE domain and the Habc domain is shortened so that it cannot span the D2-ring anymore – would such complexes be resistant to disassembly?

Thank you for pointing this out. Indeed, we have previously shown that NSF disassembles truncated SNARE complexes (<https://elifesciences.org/articles/38888>). While the D1 and D2 rings of NSF sample a split conformation, no substrate appears loaded in the D2 pore for the truncated substrate. If the linker between the SNARE and Habc domain were shortened, NSF/Sec18 could likely engage another SNARE. Indeed, in our previous work, we showed that NSF engaged with SNAP25. In more recent work (<https://www.biorxiv.org/content/10.1101/2024.10.11.617886v1>), we also observe that syntaxin can be engaged in the D1 pore of NSF. In the present work, we observe Sso1 (Syntaxin ortholog) to be engaged in the D1 pore of Sec18. Given that the interactions underlying pore loop engagement appear non-specific, Sec18/NSF likely engages the most accessible parts of a given SNARE in an unfolded conformation.

Responses to Reviewer 4

The data appear of high technical quality and the conclusions plausible. The work is accessible to a broad audience and is likely to be an important advance for the membrane fusion, vesicle trafficking, and AAA+ ATPase fields, as it provides a compelling explanation for how Sec18/NSF can unfold SNAREs in which access to a free SNARE chain is blocked by folded domains on both ends while also expanding the number of AAA+ systems that use side loading mechanisms for substrate engagement.

Pending the resolution of a few questions and issues, publication in NSMB would seem warranted.

We thank reviewer 4 for their kind words.

Lines 154-155. Please show density for all stated nucleotide assignments.

We have included the densities for the nucleotide assignments now in **Extended Data Figure 3A**.

Line 172, Section on substrate-free Sec18 structures. Why were Sec18 oligomers dissociated into monomers and then reassembled as opposed to just using native, substrate-free oligomers? How can one know that this workflow isn't giving rise to improperly assembled oligomers? Is it just because of the activity seen in ED Fig. 6H-J? Is this level of activity the same as seen for native Sec18 that hasn't been disassembled?

We thank the reviewer for the excellent question. The disassembly and reassembly protocol we used for Sec18 is a previously published protocol to prepare high-quality NSF samples (<https://pubmed.ncbi.nlm.nih.gov/25581794/>). For Sec18, we had to employ this same procedure for the substrate-free cryo-EM because we could not successfully perform cryo-EM on a more 'native' Sec18 without substrate (contaminants and aggregation made picking particles impossible after initial collection).

The Sec18 oligomers are likely properly assembled, given that they are identical to what we observe in the y20S hydrolyzing condition, in which we start with the y20S complex and then trigger hydrolysis. As the reviewer points out, this sample shows significant activity in the assays shown in **Extended Data Fig. 6I-J**. The levels are comparable to lower purity, native Sec18 preps in this assay. Still, we could not truly compare the activities of the different types of preps because the native Sec18 preparation contains contaminants that make it challenging to make an accurate 1:1 comparison.

Fig. 3D. Please show a zoomed-in view of the Sso1 density in the D2 ring.

We have included a new sub-panel with a zoomed-in view in an **Extended Data Figure 3B**, now showing only the Sso1 density within the complex

Fig. 4G and H. Why is the signal for the doubly crosslinked Habc so much weaker than for the uncrosslinked? This signal difference makes the relative fraction of disassembly comparison questionable.

The double crosslinked Habc construct was difficult to purify since the unnatural amino acid p-azido-L-phenylalanine is incorporated at not one but two different positions, resulting in low yield. The labeling occurred post-UV treatment. While the labeling efficiency was >50% for both types of constructs, we chose to normalize our input protein amounts (not our input labeling efficiency) to ensure comparable kinetics.

Because of the signal difference, we did not choose to examine the initial rates. Instead, we looked for the presence or absence of disassembly activity because if the Habc domain is threaded, crosslinking would completely ablate disassembly. Because we see complete disassembly in both cases, we feel confident that Habc crosslinking does not ablate disassembly, which is consistent with our single-molecule FRET data.

Fig. 6. What promotes ring opening? Could a closed ring engage the SNARE substrate through the N-domains, which might then trigger opening? Can open rings be seen if Sec18 is mixed with a SNAP/SNARE complex lacking the Sso1 Habc domain and either some or all of the linker?

We thank the reviewer for the excellent question. The N-domains may grab the substrate first and then cause the splitting of the D1 and D2 rings to occur to allow for loading. However, given that we observe the split ring state without a substrate, the N-domain engagement and the ring splitting are not necessarily correlated. We have also included the new **Extended Data Figure 8**, which depicts the analysis and shows what residues contribute to ring opening. In **Figure 7**, the open rings are observed with Sec18 lacking ANY substrate (no Sec17 or γ SNAREs), showing that Sec18 randomly samples this state until it encounters substrate. Please also see the new section “Mechanistic details of ring opening” (lines 343-365) and the expanded Discussion (lines 404-421).

Fig. 6. What is the role of the heptamer?

We agree with reviewer 4 that the presence of heptameric Sec18 under hydrolyzing is surprising; we were also initially surprised by this result. We have added a section in the Discussion to clarify this further (lines 404-419). Given the asymmetry of the hexameric state, it is possible that incorporating a seventh protomer resolves some overall tension in the oligomer and stabilizes the ring system under the *in vitro* conditions we explored. However, the physiological relevance of the heptamer is unclear. In the Y20S hydrolyzing condition, the split hexamer transitions to heptamer when there is no more substrate to process. However, the heptamer may not be a natural condition in the cell, as there is likely an abundance of substrate *cis*-SNARE to capture and process. We also re-examined our XL-MS data but could not find any crosslinks that could disentangle the presence of a hexamer or heptamer due to the small overall changes in residue distances between the different states. Based on this, we propose that, unless there is a significant dearth of substrate to process, NSF/Sec18 likely remain hexameric in the cell.

ED Fig. 6D. How was it confirmed from the native gel that the major band is the heptamer and the minor species a hexamer? Could an alternative method like mass photometry be used to show this?

Two bands only appear on a native gel, but one band appears on the SDS Page gel (**Extended Data Figures 6B and 6D**), indicating that the two bands on the native gel

are composed of the same protomers. The weights of these bands correspond to the hexamer and heptamer weights, but most importantly, the sample that was run on the native gel is the same sample we studied by cryo-EM, in which we found both a heptamer and hexamer.